# MIGGPT: Harnessing Large Language Models for Automated Migration of Out-of-Tree Linux Kernel Patches Across Versions

**Pucheng Dang** [1,2,3]    **Di Huang** [1]    **Dong Li** [1,2,3] *    **Kang Chen** [4]
**Yuanbo Wen** [1]    **Qi Guo** [1]    **Xing Hu** [1,3]
[1] State Key Lab of Processors, Institute of Computing Technology, CAS
[2] University of Chinese Academy of Sciences
[3] Zhongguancun Laboratory
[4] Tsinghua University
{dangpucheng20g,lidong}@ict.ac.cn

## Abstract

Out-of-tree kernel patches are essential for adapting the Linux kernel to new hardware or enabling specific functionalities. Maintaining and updating these patches across different kernel versions demands significant effort from experienced engineers. Large language models (LLMs) have shown remarkable progress across various domains, suggesting their potential for automating out-of-tree kernel patch migration. However, our findings reveal that LLMs, while promising, struggle with incomplete code context understanding and inaccurate migration point identification. In this work, we propose MIGGPT, a framework that employs a novel code fingerprint structure to retain code snippet information and incorporates three meticulously designed modules to improve the migration accuracy and efficiency of out-of-tree kernel patches. Furthermore, we establish a robust benchmark using real-world out-of-tree kernel patch projects to evaluate LLM capabilities. Evaluations show that MIGGPT significantly outperforms the direct application of vanilla LLMs, achieving an average completion rate of **74.07%** (↑ 45.92%) for migration tasks. Our code and data are available at https://github.com/CherryBlueberry/MigGPT.

## 1  Introduction

The Linux kernel, a widely-used open-source operating system, is extensively applied across various domains [46, 30, 5, 55]. Its adaptability and extensibility enable developers to create out-of-tree kernel patches that enhance performance [23, 1] or security [50, 57, 52], holding irreplaceable significance in modern engineering practices. Out-of-tree kernel patches, such as RT-PREEMPT, AUFS, HAOC, Raspberry Pi kernel, and Open vSwitch, are modifications to the Linux kernel that are developed and maintained independently of the mainline source tree. Unlike in-tree patches, which are included in official kernel releases, out-of-tree patches address specific use cases or features not yet supported by the mainline kernel. As the Linux kernel evolves, these out-of-tree patches require ongoing maintenance to ensure compatibility with newer Linux kernel versions. As shown in Figure 1, the maintenance process involves utilizing the old out-of-tree kernel patch and analyzing the differences between the old and new Linux kernel versions to upgrade the patched kernel repository to the new version. This maintenance process is crucial and labor-intensive in engineering applications, which demands specialized experts and takes weeks of intensive effort [56].

---

*Corresponding author

39th Conference on Neural Information Processing Systems (NeurIPS 2025).

Existing code migration technologies [48, 25, 12, 17, 47, 22, 41, 10, 45, 43, 44, 53] utilize static program analysis [26] to facilitate API cross-version maintenance or the backporting of CVE security patches. However, these methods only address a subset of scenarios in out-of-tree kernel patch migration. 1) They rely on pre-defined migration rules, which are insufficient for handling comprehensive scenarios involving complex changes such as namespace modifications, invocation conflict resolution, and the integration of control and data flow dependencies. 2) These single-step methods assume known target code locations in updated repositories, limiting their applicability to out-of-tree kernel patch scenarios.

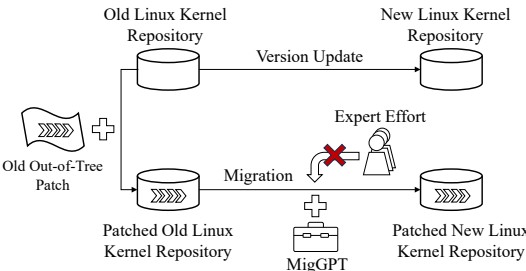

Figure 1: MIGGPT can assist in automating the version migration and maintenance of out-of-tree kernel patches of the Linux kernel. This saves on expert labor costs and reduces the development cycle.

With the substantial progress made by Large Language Models (LLMs) in understanding [38, 19, 24] and generating code [13, 2, 36, 51, 54], there is a promising opportunity to leverage LLMs for the automated migration and maintenance of out-of-tree kernel patches. However, due to the inherent lack of determinism in LLMs when generating content, several challenges arise when directly employing these models to handle the migration and maintenance of out-of-tree kernel patches. These challenges include 1) structural similarity-induced failure, 2) non-deterministic code snippet boundaries, 3) absence of associated code snippet information, and 4) inaccuracies in locating migration points, which reveal that LLMs struggle with incomplete code context understanding and inaccurate migration point identification.

To address the challenges, we propose MIGGPT, the first framework designed to assist humans in automating the migration and maintenance of out-of-tree kernel patches. We reframe this migration process as a two-step task: retrieving target code from updated kernels and performing patch migration, which is both more challenging and practical. MIGGPT utilizes Code Fingerprint (CFP), a novel data structure to encapsulate the structural and critical information of code snippets throughout the migration process of out-of-tree kernel patches. With the assistance of CFP, MIGGPT incorporates three core modules: the Retrieval Augmentation Module (addressing challenges 1 and challenges 3), the Retrieval Alignment Module (addressing challenge 2), and the Migration Enhancement Module (addressing challenge 4). Specifically, the Retrieval Augmentation Module supplies code snippet information via CFPs, mitigates interference from similar structures, and appends additional code snippet information pertinent to migration. The Retrieval Alignment Module achieves alignment of the target code snippet boundaries through the first anchor line and the last anchor line within CFPs. The Migration Enhancement Module facilitates accurate and efficient migration by comparing CFPs to ascertain the number of migration points and their respective locations.

To evaluate the efficiency of MIGGPT, we construct a robust benchmark that includes three real-world projects from the out-of-tree patch community of the Linux kernel. These projects comprise two different levels of migration examples, encompassing a variety of common migration types. With this benchmark, we evaluated MIGGPT across diverse LLMs (GPT-3.5, GPT-4-turbo, OpenAI-o1, DeepSeek-V3, DeepSeek-R1, and Llama-3.1) [33, 34, 35, 8, 6, 11] to validate its effectiveness and broad applicability. MIGGPT significantly outperforms the direct application of vanilla LLMs, achieving an average completion rate of **74.07%** (↑ 45.92%) for migration tasks. Additionally, the average number of queries to LLMs is only 2.26(↑ 0.26), indicating no substantial increase in computational overhead. Meanwhile, MIGGPT required only **2.08%** of the average time taken by human experts, demonstrating its superior time efficiency.

In summary, we make the following contributions:

- We have developed a robust migration benchmark, encompassing three real-world projects. To the best of our knowledge, this is the first benchmark for out-of-tree kernel patch migration that can assess performance across diverse migration tools, providing a valuable foundation for future research.

- We propose CFP, a carefully designed data structure that encapsulates the structural and critical information of code snippets, providing essential migration context for LLMs. Based

on this, we introduce MIGGPT, a framework to assist humans in automating out-of-tree kernel patch migration and maintenance.

- We conduct comprehensive experiments on both closed-source models (*i.e.* GPT-3.5, GPT-4 and OpenAI-o1) and open-source models (*i.e.* DeepSeek-V3, Deepseek-R1, and Llama-3.1). The results demonstrate that MIGGPT achieved an average migration accuracy of $74.07\%(\uparrow 45.92\%)$, representing a significant improvement over vanilla LLMs.

## 2 Related Work

### 2.1 Code Migration Kernel Patch

Existing code migration [9, 37] technologies primarily focus on API cross-version maintenance [48, 25, 12, 17, 47, 22, 41, 10] and the backporting of CVE security patches [45, 43, 44, 53]. The most similar migration efforts, FixMorph [43], TSBPORT [53], and PPatHF [37], focus on CVE patch or forked code, which **cannot be applied to the migration of out-of-tree kernel patch**: 1) these methods only partially address out-of-tree kernel patch migration due to the tight coupling between kernel and patch code. They identify vulnerability patterns and apply predefined rules [26] but fail to manage complex changes such as namespace modifications, invocation conflicts, and control/data flow dependencies, limiting their effectiveness in comprehensive migration scenarios. 2) These works handle single-step migration, assuming known target code locations in updated repositories, making them difficult to apply to out-of-tree kernel patch scenarios. In contrast, our MIGGPT tackles a more complex two-step task: retrieving target code from updated kernels and migrating patches, which is both more challenging and more practical.

### 2.2 LLMs for Coding

In recent years, LLMs [4, 14, 42, 27, 32, 28, 34] have achieved remarkable progress in various natural language processing tasks. Initially focused on natural language understanding and generation, the adaptability of LLMs has expanded to the field of software engineering, where they can be fine-tuned to perform programming tasks such as code completion [38, 19, 24], code search [13], code summarization [2], code generation [36], and even complex code repair [15, 20]. This inspires us to apply LLMs to the migration of out-of-tree kernel patches. To the best of our knowledge, MIGGPT is the first work to apply LLMs to this task, paving the way for subsequent research.

## 3 Problem Formulation

Out-of-tree kernel patches lack official support and require manual maintenance to ensure compatibility with future Linux kernel versions. An example of migration is provided in App. B. Let $R$ denote a Linux kernel repository, where $s \in R$ represents a code snippet within the repository. The older version of Linux kernel is $R_{\text{old}}$, and after applying an out-of-tree patch, it becomes $R'_{\text{old}}$. When the kernel advances to a new version $R_{\text{new}}$, the migration problem is to construct a function $M : R'_{\text{old}} \to R'_{\text{new}}$ where

Table 1: Formalization and counts of the two types of migration examples. Other cases are too simple to necessitate resolution. Details in App. A.2.

| Class | Formalization | Number |
|---|---|---|
| Type 1 | $\Delta \neq \varnothing,\ \Sigma \neq \varnothing,$ $\forall \delta \in \Delta,\ \forall \sigma \in \Sigma,\ \langle \delta, \sigma \rangle = 0$ | 80 (59.3%) |
| Type 2 | $\Delta \neq \varnothing,\ \Sigma \neq \varnothing,$ $\forall \delta \in \Delta,\ \exists \sigma \in \Sigma,\ \langle \delta, \sigma \rangle \neq 0$ | 55 (40.7%) |
| Others | $\Delta = \varnothing$ or $\Delta \neq \varnothing,\ \Sigma = \varnothing$ | Too simple to resolve |

$\forall s \in R'_{\text{old}}, \exists M(s) \in R'_{\text{new}}$ s.t.$\forall x \in \text{Inputs}, \text{Execute}(R'_{\text{old}}, x) = \text{Execute}(R'_{\text{new}}, x)$.

## 4 Migration Benchmark

### 4.1 Migration Types

We can obtain the code snippets $s_{\text{old}} \in R_{\text{old}}$ and $s'_{\text{old}} \in R'_{\text{old}}$ at the same location in the repository before and after applying the out-of-tree kernel patch, with the differences represented by $\Delta$. As $R_{\text{old}}$ is updated to a new version of the Linux kernel $R_{\text{new}}$, we need to locate the corresponding code snippet $s_{\text{new}} \in R_{\text{new}}$ in the new version of the Linux kernel to obtain the difference information

during the kernel update. The differences between $s_{\text{old}}$ and $s_{\text{new}}$ are denoted as $\Sigma$. Subsequently, by utilizing the information from $\Delta$ and $\Sigma$, we complete the migration task to obtain the new version of the out-of-tree kernel patch code snippet $s'_{\text{new}}$. Finally, these code snippets are integrated to form the new version of the out-of-tree kernel patch code repository $R'_{\text{new}}$.

Considering the states of $\Delta$ and $\Sigma$, we can categorize the migration types into two classes:

**Type 1**: This type of migration example satisfies $\Delta \neq \varnothing, \Sigma \neq \varnothing, \forall \delta \in \Delta, \forall \sigma \in \Sigma, \langle \delta, \sigma \rangle = 0$. This indicates that both the out-of-tree kernel patch and the new version of the Linux kernel have modified the code snippet, and their changes do not affect the same lines of code, meaning the modifications do not overlap or conflict with each other. Traditional methods' limitations with Type 1 are discussed in App. G.1.

**Type 2**: In contrast, this type satisfies $\Delta \neq \varnothing, \Sigma \neq \varnothing, \forall \delta \in \Delta, \exists \sigma \in \Sigma, \langle \delta, \sigma \rangle \neq 0$, indicating that their modifications overlap on the same lines of code, leading to conflicts.

The remaining cases, $\Delta = \varnothing$ and $\Delta \neq \varnothing, \Sigma = \varnothing$, signify no code modification in the out-of-tree kernel patch and no changes in the new kernel version, respectively. Due to their simplicity and straightforward migration, they are excluded from our benchmark.

## 4.2 Benchmark Design

We have built a robust migration testing benchmark using out-of-tree kernel patches from real-world projects, specifically focusing on three open-source initiatives: RT-PREEMPT [21], Raspberry Pi Linux [39], and HAOC [16] [2]. More details about these projects are available in App. A.1. We collect code from these projects across Linux kernel versions 4.19, 5.4, 5.10, and 6.6 for our benchmark. The selected kernel versions are officially maintained LTS releases, which are widely adopted in production systems(e.g., enterprise servers, embedded devices).

Guided by the experience of manually completing the task, we divide the migration task into two steps: 1) Identifying the migration location, i.e., finding $s_{\text{new}}$. 2) Completing the migration to obtain $s'_{\text{new}}$. In this case, firstly, we use the `diff` command to obtain the code snippets $s_{\text{old}}$ and $s'_{\text{old}}$ from files with the same name in the code repository. Subsequently, by matching filenames, we locate the file in code repository $R_{\text{new}}$ that contains the target new version code snippet $s_{\text{new}}$. Finally, we gather the ground truth (results manually completed by humans) $\hat{s}_{\text{new}}$ and $\hat{s}'_{\text{new}}$. Specifically, our benchmark includes a quintuple $(s_{\text{old}}, s'_{\text{old}}, \text{file}_{\text{new}}, \hat{s}_{\text{new}}, \hat{s}'_{\text{new}})$ for each migration example. After filtering out invalid differences (such as spaces, blank lines, file deletions, etc.), we randomly collected 135 migration examples, comprising 80 Type 1 and 55 Type 2, as detailed in Table 1.

## 5 MIGGPT

We first outline the challenges faced when utilizing vanilla LLMs for the migration of out-of-tree kernel patches (Section 5.1), and then discuss how MIGGPT effectively addresses these challenges (Sections 5.2 to 5.7).

## 5.1 Challenges

Through analyzing LLM behavior and results, we identify key challenges hindering their success in out-of-tree kernel patch migration:

**Challenge 1 (Structural Ambiguity)**: When identifying the code snippet $s_{\text{new}}$ in $\text{file}_{\text{new}}$ for migration, retrieval errors can occur. LLMs often struggle to locate function definitions within $s_{\text{new}}$ due to interference from similar function structures, leading to inaccuracies that affect subsequent migration stages. An example is provided in App. C.1.

**Challenge 2 (Boundary Indeterminacy)**: This challenge occurs when retrieving $s_{\text{new}}$ from $\text{file}_{\text{new}}$. Due to the inherent randomness in LLM-generated responses, discrepancies often arise between the start and end lines of $s_{\text{new}}$ identified by the LLM and those retrieved by human developers ($\hat{s}_{\text{new}}$).

---

[2]Even with knowledge of the code in these out-of-tree kernel patches, LLMs still struggle to accomplish migration and maintenance tasks.

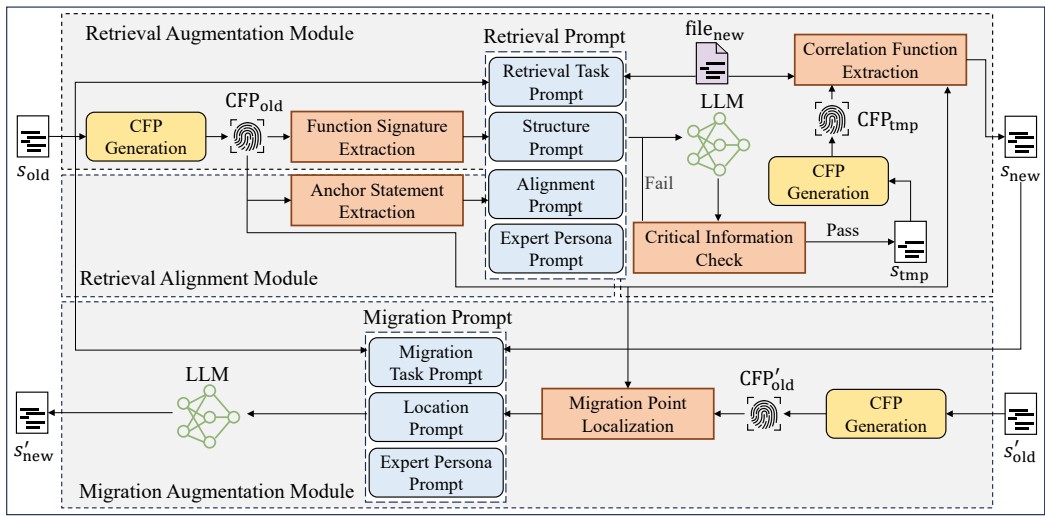

Figure 2: Overview of MIGGPT. MIGGPT employs a code fingerprint (CFP) structure to retain code snippet information, enhanced by three modules to improve migration accuracy and efficiency. The migration process involves two steps: 1) locating the migration position in file$_{new}$ to find $s_{new}$, and 2) completing the migration to obtain $s'_{new}$.

This indeterminacy can result in missing or extraneous lines, significantly compromising migration outcomes where precise code segment boundaries are critical. An example is provided in App. C.2.

**Challenge 3 (Missing Associated Fragments)**: This challenge occurs when retrieving $s_{new}$ from file$_{new}$. During Linux kernel upgrades, code blocks from older versions may be split into fragments in the new version for standardization or reuse. LLMs often fail to identify and retrieve all these fragments, leading to incomplete $s_{new}$. This results in errors during out-of-tree kernel patch migration due to missing code segments. An example is provided in App. C.3.

**Challenge 4 (Ambiguous Migration Points)**: This challenge arises during the migration of $s_{new}$ to $s'_{new}$. Although the information provided by $s_{old}$ and $s'_{old}$ is sufficient to accurately infer the migration point, LLMs frequently fail to precisely identify these points. This ambiguity results in errors when determining the correct location for migration. An example is provided in App. C.4.

Overall, LLMs require migration-relevant code structure information and code scope constraints to more effectively migrate and maintain out-of-tree kernel patches.

## 5.2 Overview

## 5.3 Code Fingerprint

To this end, we propose MIGGPT, a framework combining traditional program analysis with LLMs to facilitate out-of-tree kernel patch migration across Linux versions. As outlined in Section 4, MIGGPT works in two stages: identifying target code snippets in the new version and migrating the out-of-tree patch. Figure 2 shows its three core modules: the **Retrieval Augmentation Module** (addressing Challenges 1 and 3), the **Retrieval Alignment Module** (addressing Challenge 2), and the **Migration Enhancement Module** (addressing Challenge 4). Each module uses a code fingerprint structure, which encodes the structural features of code snippets, to enhance LLM performance and migration accuracy, tackling the challenges discussed earlier.

```
1   static inline void local_daif_mask(int set_mm)
2   {
3       asm volatile("msr daifset, #0xf"...);
4       if (system_uses_nmi())
5           _allint_set();
6       /* Don't really care for a dsb here */
7       trace_hardirqs_off();
8       ...
9   }
```

Figure 3: A code snippet containing inline assembly statements and comment annotations.

To address the challenges LLMs face in migrating out-of-tree kernel patches across Linux kernel versions, a detailed analysis of code snippet structure is essential to identify migration-related code structure information and code scope constraints. While tools like Abstract Syntax Tree (AST) are useful for structural analysis, they have limitations: 1) Inability to process code snippets that lack

complete compilation dependencies (e.g., missing variable or function definitions, absent macro declarations, or incomplete header inclusions) due to tight integration with the compilation process. 2) The mismatch between excessive structural details (AST tools provide a plethora of information irrelevant to patch migration) and the absence of critical information (such as comments and inline assembly), which is essential for maintaining and updating kernel patches [3]. Focusing on key statements, such as migration points and alignment positions, while preserving essential elements like comments and inline assembly, can enhance efficiency and reduce overhead in the migration process.

To address the limitations of traditional code structure analysis, we propose Code Fingerprint (CFP), a lightweight sequential data structure for analyzing code snippets. CFP records both the content and positional information for each line of statements, encompassing all C language statements, including comments and inline assembly (a detailed example is provided in App. D.1). As shown in Figure 4, CFP focuses on recording function definitions and function calls, which are crucial for addressing challenges 1 and 3, as detailed in Section 5.4. Additionally, its linear structure facilitates accurate positioning for insertion, deletion, and other update operations, tackling challenges 2 and 4, further explained in Sections 5.5 and 5.6. The algorithm for generating CFP is in App. D.5. Overall, CFP offers three key advantages: 1) effective processing of incomplete code snippets, 2) preservation of critical information such as comments and inline assembly, which are vital

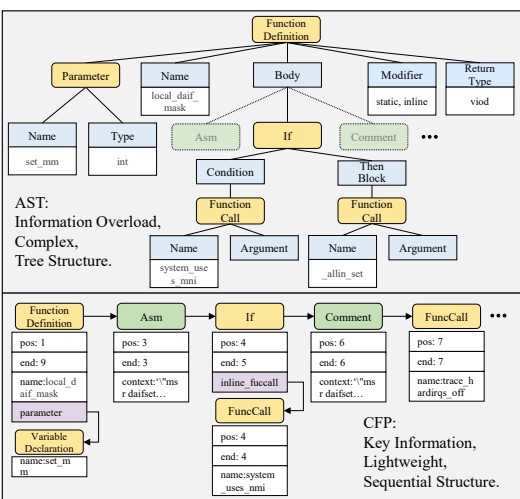

Figure 4: Compared to AST, CFP extracts key code structures, and its linear representation enables clearer localization of code modification points.

for out-of-tree kernel patch migration, and 3) a streamlined design that focuses on essential statements, improving migration efficiency and reducing overhead. CFP is specifically designed for out-of-tree kernel patch code, addressing the challenges encountered in Linux kernel patches. More discussions about CFP are in App. G.2. By minimizing unnecessary processing while ensuring relevance, CFP provides a targeted solution for migrating out-of-tree kernel patches across Linux kernel versions.

## 5.4 Retrieval Augmentation Module

The retrieval augmentation module is designed to address challenge 1 and challenge 3 encountered during the migration update of out-of-tree kernel patches by LLMs. In challenge 1, LLMs are prone to be misled by similar function structure when processing function definitions in code snippets, leading to incorrect retrieval of $s_{new}$ in $file_{new}$, which ultimately results in erroneous migrated $s'_{new}$. To overcome this challenge, it is necessary to constrain the LLM's attention to the target code snippet. As illustrated in Figure 2, the retrieval augmentation module achieves this by constructing a code fingerprint structure ($CFP_{old}$) for the old version of the Linux kernel code snippets $s_{old}$. By analyzing $CFP_{old}$, the module extracts the function signatures of the function definitions contained within $s_{old}$. These function signatures are then used to build a prompt to describe the structure information ("Structure Prompt"), which is incorporated into the input fed to the LLM. An example is provided in the App. D.2.

On the other hand, challenge 3 highlights that during the migration update of out-of-tree kernel patches by LLMs, there is an issue with missing associated functions. For the LLM's temporary retrieval result $s_{tmp}$, we utilize the code fingerprint structures $CFP_{tmp}$ and $CFP_{old}$ of $s_{tmp}$ and $s_{old}$, respectively, and extract from them the sets of internally called associated functions, denoted as $\mathcal{F}_{tmp}$ and $\mathcal{F}_{old}$. Then, using string matching techniques, we retrieve from $file_{new}$ the code snippets corresponding to the associated function calls Funccall that satisfy Funccall $\in \mathcal{F}_{tmp} \setminus \mathcal{F}_{old}$. Ultimately, these associated function code snippets are combined with $s_{tmp}$ to form the complete code snippet $s_{new}$. An example is provided in the App. D.2.

---

[3]Inline assembly is widely used in the Linux kernel, and comments are crucial for future module development, as their omission would hinder subsequent modifications.

## 5.5 Retrieval Alignment Module

The retrieval alignment module is devised to tackle challenge 2, which was encountered during the migration update of out-of-tree kernel patches by LLMs. Challenge 2 indicates that when the LLM retrieves the target code snippet $s_{\text{new}}$ from the new version of the Linux kernel file file$_{\text{new}}$, there can be a mismatch between the boundary line of $s_{\text{new}}$. To address this issue, we need to leverage the information from the first and last lines of the old version code snippet $s_{\text{old}}$ to aid in the localization during the retrieval of $s_{\text{new}}$. As illustrated in Figure 2, we utilize the code fingerprint structure CFP$_{\text{old}}$ of $s_{\text{old}}$. By taking advantage of its linear structure, we obtain the CFP statements for the first and last lines. These statements are used to construct an "Alignment Prompt", which describes the information of the first and last lines and is included as part of the input to the LLM. This prompt guides the LLM in performing the retrieval task better by accurately identifying the boundaries of the code snippet.

## 5.6 Migration Augmentation Module

The migration augmentation module is primarily designed to address challenge 4 encountered by LLMs during the migration of new-version Linux kernel code snippets $s_{\text{new}}$ into the final updated out-of-tree kernel patch $s'_{\text{new}}$. In challenge 4, LLMs often struggle to accurately identify the number and location of migration points, leading to errors in the final migrated $s'_{\text{new}}$. As illustrated in Figure 2, to tackle this challenge, we leverage information from the old version code snippet $s_{\text{old}}$ and its modified counterpart $s'_{\text{old}}$ to determine the number and location of modifications made to the out-of-tree kernel patch. This information is used to construct a "Location Prompt" that assists the LLM in accurately identifying the number and location of migration points. An example is provided in the App. D.3.

## 5.7 Implementation

With the critical code information provided by CFP, we can leverage the Retrieval Augmentation Module and the Retrieval Alignment Module to assist LLMs in more effectively identifying target kernel code snippets $s_{\text{new}}$. Subsequently, with the aid of the Migration Augmentation Module, we facilitate the migration to generate the final code snippet $s'_{\text{new}}$. All the prompts and the algorithm of MIGGPT are provided in App. D.4.

As illustrated in Figure 2, we first need to retrieve code snippet $s_{\text{new}}$ from file$_{new}$. Specifically, using the information contained within CFP$_{old}$, we can extract a set of critical function signatures $\mathcal{S}$ and a set of key anchor statements $\mathcal{A}$. With this information, we construct the $StructurePrompt$ and $AlignmentPrompt$, ultimately forming the complete $RetrievalPrompt$. We then query LLMs using the $RetrievalPrompt$ to obtain an initial result $s_{\text{tmp}}$. We check if $s_{\text{tmp}}$ contains items from the target function signature set $\mathcal{S}$. If not, we repeatedly query the LLMs using the $RetrievalPrompt$ (up to $m$ times). If it does contain items from $\mathcal{S}$, we use CFP$_{old}$ and CFP$_{tmp}$ to extract newly appeared called functions within $s_{\text{tmp}}$ and retrieve the code snippets where these called functions are defined from file$_{new}$ as additional supplementary information for $s_{\text{tmp}}$. Finally, we concatenate these two parts of the code snippets to obtain $s_{\text{new}}$. After obtaining $s_{\text{new}}$, we proceed to migrate it to achieve $s'_{\text{new}}$. We utilize the differences between CFP$_{old}$ and CFP$'_{old}$ to extract the number and positions of migration points and generate the $LocationPrompt$. Further, we formulate the $MigrationPrompt$ and query the LLM to obtain the migrated out-of-tree kernel patch code snippet $s'_{\text{new}}$.

## 6 Evaluation

In this section, we assess the performance of MIGGPT, focusing on the following questions:
**RQ1 (Performance)**: How does the performance of MIGGPT compare with that of vanilla LLM?
**RQ2 (Ablation)**: How does each module within MIGGPT contribute to the overall performance?
**RQ3 (Failure Analysis)**: How much modification is required for MIGGPT's failed example to align with human-level performance in out-of-tree patch migration tasks?

### 6.1 Evaluation Settings

We assess MIGGPT using two benchmarks: the out-of-tree kernel patch migration benchmark from Section 4 and FixMorph's CVE patch backporting benchmark [43], which includes 350 instances. For baselines, we use vanilla LLMs, including GPT-3.5 [33], GPT-4-turbo [34], OpenAI-

Table 2: The accuracy of the MIGGPT-augmented LLMs compared to vanilla LLMs on retrieving target code snippets.

| LLM | Method | Type 1 (80) | | | Type 2 (55) | | | All (135) | | | Average Query Times |
| --- | --- | --- | --- | --- | --- | --- | --- | --- | --- | --- | --- |
| | | Best Match | Semantic Match | Human Match | Best Match | Semantic Match | Human Match | Best Match | Semantic Match | Human Match | |
| GPT-3.5 | Vanilla | 20.00% | 33.75% | 26.25% | 20.00% | 25.45% | 27.27% | 20.00% | 30.37% | 26.67% | 1.00 |
| | MIGGPT | 68.75% | 68.75% | 71.25% | 61.82% | 54.55% | 70.91% | 65.93% | 62.96% | 71.11% | 1.28 |
| GPT-4-turbo | Vanilla | 60.00% | 67.50% | 65.00% | 69.09% | 76.36% | 78.18% | 63.70% | 71.11% | 70.37% | 1.00 |
| | MIGGPT | 91.25% | 95.00% | 96.25% | 81.82% | 83.64% | 89.09% | 87.41% | 90.37% | 93.33% | 1.16 |
| OpenAI-o1 | Vanilla | 76.25% | 82.50% | 80.00% | 81.82% | 85.45% | 92.73% | 78.52% | 83.70% | 85.19% | 1.00 |
| | MIGGPT | 96.25% | 97.50% | 96.25% | 85.45% | 89.09% | 92.73% | 91.85% | 94.07% | 94.81% | 1.25 |
| DeepSeek-V3 | Vanilla | 68.75% | 71.25% | 72.50% | 74.55% | 78.18% | 78.18% | 71.11% | 74.07% | 74.81% | 1.00 |
| | MIGGPT | 92.50% | 93.75% | 95.00% | 85.45% | 78.18% | 89.09% | 89.63% | 87.41% | 92.59% | 1.22 |
| DeepSeek-R1 | Vanilla | 72.50% | 76.25% | 77.50% | 63.64% | 70.91% | 74.55% | 68.89% | 74.07% | 76.30% | 1.00 |
| | MIGGPT | 95.00% | 95.00% | 95.00% | 80.00% | 85.45% | 87.27% | 88.89% | 91.11% | 91.85% | 1.23 |
| Llama-3.1-8B | Vanilla | 37.50% | 46.25% | 43.75% | 43.64% | 47.27% | 45.45% | 40.00% | 46.67% | 44.44% | 1.00 |
| | MIGGPT | 77.50% | 80.00% | 81.25% | 70.91% | 74.55% | 78.18% | 74.81% | 77.78% | 80.00% | 1.36 |
| Llama-3.1-70B | Vanilla | 58.75% | 65.00% | 63.75% | 61.82% | 72.73% | 75.55% | 60.00% | 68.15% | 68.15% | 1.00 |
| | MIGGPT | 91.25% | 92.50% | 93.75% | 80.00% | 81.82% | 81.82% | 86.67% | 88.15% | 88.89% | 1.29 |
| Average | Vanilla | 56.25% | 63.26% | 61.25% | 59.22% | 65.19% | 67.42% | 57.46% | 64.02% | 63.70% | 1.00 |
| | MIGGPT | 87.50% | 88.93% | 76.25% | 77.92% | 78.18% | 84.16% | 83.60% | 84.55% | 87.51% | 1.26 |
| | ↑ | +31.25% | +25.67% | +15.00% | +18.70% | +12.99% | +16.74% | +26.14% | +20.53% | +23.81% | - |

Table 3: The accuracy of the MIGGPT-augmented LLMs compared to vanilla LLMs on the migration task of target code snippets.

| LLM | Method | Type 1 (80) | | | Type 2 (55) | | | All (135) | | |
| --- | --- | --- | --- | --- | --- | --- | --- | --- | --- | --- |
| | | Best Match | Semantic Match | Human Match | Best Match | Semantic Match | Human Match | Best Match | Semantic Match | Human Match |
| GPT-3.5 | Vanilla | 7.50% | 5.00% | 8.75% | 3.64% | 3.64% | 5.45% | 5.93% | 4.44% | 7.41% |
| | MIGGPT | 37.50% | 46.26% | 47.50% | 38.18% | 41.82% | 61.82% | 37.78% | 44.44% | 53.33% |
| GPT-4-turbo | Vanilla | 15.00% | 12.50% | 18.75% | 10.91% | 30.91% | 23.64% | 13.33% | 20.00% | 20.74% |
| | MIGGPT | 68.75% | 82.50% | 85.00% | 54.55% | 76.36% | 69.09% | 62.96% | 80.00% | 78.52% |
| OpenAI-o1 | Vanilla | 20.00% | 28.75% | 30.00% | 18.18% | 30.91% | 27.27% | 19.26% | 29.63% | 28.89% |
| | MIGGPT | 77.50% | 90.00% | 90.00% | 60.00% | 76.36% | 74.55% | 70.37% | 84.44% | 83.70% |
| DeepSeek-V3 | Vanilla | 23.75% | 37.50% | 32.50% | 34.55% | 54.55% | 49.09% | 28.15% | 44.44% | 39.26% |
| | MIGGPT | 81.25% | 88.75% | 87.50% | 65.45% | 78.18% | 74.55% | 74.81% | 84.44% | 82.22% |
| DeepSeek-R1 | Vanilla | 53.75% | 60.00% | 62.50% | 40.00% | 54.55% | 56.36% | 48.15% | 57.78% | 60.00% |
| | MIGGPT | 72.50% | 85.00% | 81.25% | 69.09% | 81.82% | 78.18% | 71.11% | 83.70% | 80.00% |
| Llama-3.1-8B | Vanilla | 5.00% | 12.50% | 16.25% | 0% | 20.00% | 25.45% | 2.96% | 15.56% | 20.00% |
| | MIGGPT | 36.25% | 65.00% | 67.50% | 25.45% | 52.73% | 56.36% | 31.85% | 60.00% | 62.96% |
| Llama-3.1-70B | Vanilla | 3.75% | 16.25% | 18.75% | 7.27% | 27.27% | 23.64% | 5.19% | 20.74% | 20.74% |
| | MIGGPT | 62.50% | 80.00% | 81.25% | 47.27% | 67.27% | 72.73% | 56.30% | 74.81% | 77.78% |
| Average | Vanilla | 18.39% | 24.64% | 26.79% | 16.36% | 31.69% | 30.13% | 17.57% | 27.51% | 28.15% |
| | MIGGPT | 62.32% | 76.78% | 77.14% | 51.43% | 67.79% | 69.61% | 57.88% | 73.12% | 74.07% |
| | ↑ | +43.93% | +52.14% | +50.36% | +35.06% | +36.10% | +39.48% | +40.32% | +45.61% | +45.92% |

o1 [35], DeepSeek-V2.5 [7], DeepSeek-V3 [8], Deepseek-R1 [6], Llama-3.1-8B and Llama-3.1-70B-Instruct [11], as they are widely recognized for their advanced performance, along with previous migration efforts like FixMorph [43], TSBPORT [53] and PPatHF [37]. Evaluation metrics include "best match" (exact code similarity after removing spaces, line breaks, and tab characters), "semantic match" (CodeBLEU with a 0.9 threshold for binary classification, detailed in App. F.6) [40], and "human match" (developer-judged functional equivalence, detailed in App. F.3). We also considered the compilation success rate, as detailed in App. F.4. The hyperparameter $m$ is set to 3.

For each sample $(s_{old}, s'_{old}, file_{new}, \hat{s}_{new}, \hat{s}'_{new})$ in our benchmark, we evaluate vanilla LLMs using two distinct strategies: **One-step Strategy**: The LLM directly generates the migrated code snippet $s'_{new}$ by taking the triplet $(s_{old}, s'_{old}, file_{new})$ as input. **Two-step Strategy**: The process is divided into two phases. First, the LLM identifies the corresponding new version code snippet $s_{new}$ using the pair $(s_{old}, file_{new})$. Then, the LLM generates $s'_{new}$ by taking the triplet $(s_{old}, s'_{old}, s_{new})$ as input.

## 6.2 Performance Evaluation (RQ1)

**MIGGPT demonstrates exceptional capability in retrieving target code snippets.** As shown in Table 2, MIGGPT exhibits a significant advantage in the subtask of retrieving target code. Specifically, when paired with a high-performance LLM like GPT-4-turbo, MIGGPT achieves a human matching precision of 96.25% for Type 1 samples, significantly outperforming standalone GPT-4-turbo (65.00%). Overall, MIGGPT attains an average semantic matching precision of 84.55% across all sample types, marking a 20.53% relative improvement.

**MIGGPT demonstrates outstanding performance in generating migrated code snippets.** As shown in Table 3, MIGGPT outperforms

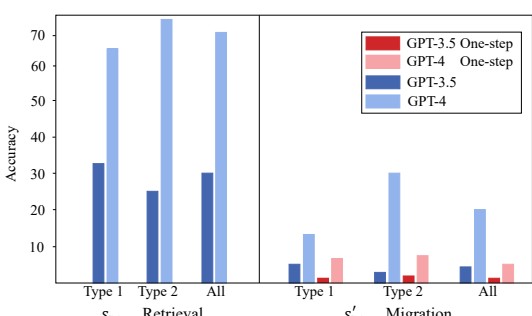

Figure 5: The semantic match accuracy of the target code snippets retrieval task and the target code snippets migration task across various LLMs. "One-step" indicates the direct utilization of an LLM to complete the migration task in a single step.

vanilla LLMs, achieving a 73.12% higher average migration semantic matching precision, a 45.61% relative improvement. Notably, due to Llama-3.1-8B's weak base performance, MigGPT-augmented

Llama-3.1-8B underperforms–yet still achieves 44.44% higher semantic matching precision than vanilla Llama-3.1-8B. Additional results for other LLMs are provided in App. E.

**The two-step strategy outperforms the one-step strategy.** We compared GPT-3.5 and GPT-4-turbo using both one-step and two-step strategies to investigate the impact of task complexity on migration performance. As illustrated in Figure 5, when vanilla LLMs are employed, the two-step strategy achieves an average migration accuracy of 12.22% across all sample types, representing an improvement of 8.89% over the one-step strategy's accuracy of 3.33%.

**MIGGPT performs well on compilation-level metric.** We conducted additional compilation success rate experiments on migrated patches. As shown in Table 4, MigGPT achieves a 48.49% average improvement in compilation success rate over vanilla LLMs.

Table 4: The compilation success rate MIGGPT compared to vanilla LLMs.

| Method | GPT-4-turbo | | DeepSeek-V3 | |
|---|---|---|---|---|
| | Vanilla | MIGGPT | Vanilla | MIGGPT |
| Rate | 16.30% | 66.67% (+50.37%) | 31.85% | 79.26% (+47.41%) |

**MIGGPT outperformed previous migration efforts on the CVE patch backporting task.** We also evaluate the performance of MIGGPT in the context of CVE patch backporting. Notably, our out-of-tree patch migration task is

Table 5: The semantic match accuracy of MIGGPT compared to patch backporting methods.

| Method | FIXMORPH | TSBPORT | PPatHF | GPT-4-turbo | | DeepSeek-V3 | |
|---|---|---|---|---|---|---|---|
| | | | | vanilla | MIGGPT | vanilla | MIGGPT |
| Accuracy | 24.63% | 87.59% | 75.12% | 85.43% | 91.78% | 87.12% | **92.59%** |

different from FixMorph, TSBPORT, and PPatHF, which only solve the migration problem, while we target both target code retrieval and migration. As illustrated in Table 5, MIGGPT demonstrates superior performance compared to previous patch backporting methods.

**MIGGPT performs well on the Linux driver migration task.** We conducted experiments to assess MIGGPT's performance on the Linux driver migration. We randomly collected 24 driver migration samples. The results, presented in the Table 6, indicate that MIGGPT performs well in migrating patches involving driver interface modifications.

Table 6: The accuracy of MIGGPT compared to vanilla LLM on the Linux driver migration task.

| Method | GPT-4-turbo | | DeepSeek-V3 | |
|---|---|---|---|---|
| | Vanilla | MIGGPT | Vanilla | MIGGPT |
| Best Match | 54.16% | 83.33% (+29.17%) | 58.33% | 79.16% (+20.83%) |
| Semantic Match | 62.50% | 83.33% (+20.83%) | 62.50% | 87.50% (+25.00%) |

**MIGGPT demonstrates cross-language generalizability.** We conducted an experiment focusing on Python code migration. We randomly selected 31 migration examples from our benchmark and translated them to Python. With only minor adjustments (modify the implementation

Table 7: The accuracy of MIGGPT compared to vanilla LLM on Python code migration task.

| Method | GPT-4-turbo | | DeepSeek-V3 | |
|---|---|---|---|---|
| | Vanilla | MIGGPT | Vanilla | MIGGPT |
| Best Match | 61.29% | 80.65% (+19.36%) | 64.52% | 83.87% (+19.35%) |
| Semantic Match | 70.97% | 87.10% (+16.13%) | 74.19% | 90.32% (+16.13%) |

of the statement tokenization and statement element extraction functions to adapt the CFP for Python syntax) MIGGPT can process these Python examples. The results of Table 7 demonstrate that the average semantic match performance improved by 16.13%. compared to a vanilla approach, indicating MIGGPT's effectiveness beyond C-language code. We also discuss the generalization of MIGGPT in App. G.3.

**MIGGPT is more time-efficient.** We compared MIGGPT with three human experts on our out-of-tree patch migration benchmark. As shown in Table 8, MIGGPT required only **2.08%** of the average time taken by human experts, demonstrating its superior time efficiency.

Table 8: The time cost of MIGGPT compared to human experts.

| Method | Expert A | Expert B | Expert C | MIGGPT | |
|---|---|---|---|---|---|
| | | | | GPT-4-turbo | DeepSeek-V3 |
| Time (days) | 14.15 | 10.89 | 12.51 | 0.25 | 0.27 |

## 6.3 Ablation Study (RQ2)

## 6.4 Ablation Study

We conduct an ablation study to evaluate the impact of the four units in MIGGPT: CFP, Retrieval Augmentation Module, Retrieval Alignment Module, and Migration Augmentation Module (details in App. F.7). Figure 6 presents the outcomes of four tested variants on our benchmarks. Among these, MIGGPT consistently outperforms the ablation baselines. Meanwhile, we perform an ablation study on the hyperparameter $m$ in Algorithm 1 with MigGPT-GPT-4-turbo, which controls the total query time of the Retrieval Augmentation Module. As shown in Figure 6, $m = 3$ is suitable for both Type 1 and Type 2 examples.

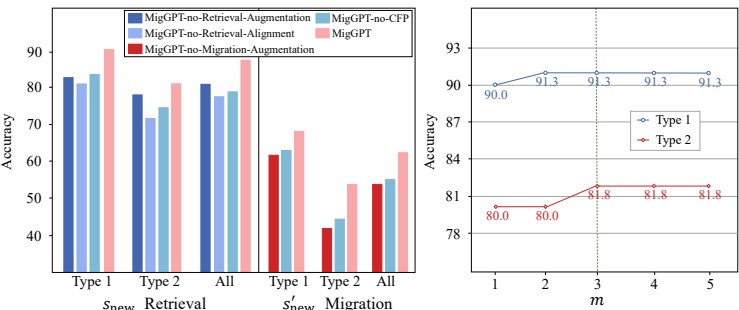

Figure 6: *Left*: The accuracy of different variants of MIGGPT. *Right*: The best match retrieval accuracy of different $m$.

## 6.5 Failure Analysis (RQ3)

We evaluate MIGGPT's robustness by analyzing failed migration cases (not human-matched) across various samples, measuring line edit distances (insertions, deletions, modifications) between MIGGPT's incorrect outputs and human-corrected results (see App. F.5). As shown in Table 9, **41% of MIGGPT's errors require fewer than three lines of modification to align with correct results**, demonstrating its potential to aid in out-of-tree kernel patch migration.

Table 9: The line edit distance between the failure cases of MIGGPT-augmented GPT-4-Turbo and the manual migration results. "$3 \leq$ dis $< 6$" denotes a line edit distance of at least 3 but less than 6.

| LLM | Type | dis < 3 | 3 ≤ dis < 6 | 6 ≤ dis < 9 | 9 ≤ dis | All |
|---|---|---|---|---|---|---|
| GPT-3.5 | Type 1 | 13 | 9 | 8 | 12 | 42 |
| | Type 2 | 8 | 4 | 2 | 7 | 21 |
| GPT-4-turbo | Type 1 | 5 | 1 | 3 | 3 | 12 |
| | Type 2 | 9 | 1 | 0 | 7 | 17 |
| DeepSeek-V2.5 | Type 1 | 10 | 2 | 3 | 1 | 16 |
| | Type 2 | 8 | 1 | 3 | 4 | 16 |
| DeepSeek-V3 | Type 1 | 3 | 2 | 3 | 2 | 10 |
| | Type 2 | 5 | 4 | 1 | 4 | 14 |

In our in-depth analysis of "human match" errors in MIGGPT-augmented GPT-4-turbo, we identified the following primary categories:

- Incomplete $s_{new}$ Retrieval for Large Codebases (31.03%): In some instances, when retrieving $s_{new}$ from file$_{new}$, the target code $s_{new}$ was significantly large (exceeding 150 lines), leading to incomplete retrieval. This could be due to LLMs' tendency to shift attention when dealing with long contexts.

- Deviation from the Migration Point in $s'_{new}$ Generation (24.14%): Even with precise migration point information provided by CFP, the LLM occasionally failed to strictly adhere to these locations during $s'_{new}$ generation. While minor offsets often didn't impact functionality, they sometimes led to functional errors in code with complex control or data flows. This behavior appears to be related to the inherent randomness of LLMs.

- Difficulty in Fusing Divergent Changes in Type 2 Migrations (27.59%): For certain Type 2 migrations, significant differences between $s_{new}$ and $s_{old}$'s modifications to $s_{old}$ prevented the LLM from correctly integrating these changes, resulting in errors. This limitation points to challenges related to the LLM's code comprehension and manipulation capabilities.

- Miscellaneous (17.24%): Errors in symbols, statements, etc., appearing in $s'_{new}$, such as incorrectly writing `verbose(env, off, size, reg's id)` instead of `verbose(env, off, size, reg->id)`. This may be related to LLM hallucinations.

It's important to note that these identified failure modes primarily arise from the inherent limitations of the LLMs themselves, rather than architectural flaws within the MigGPT framework.

## 7 Conclusion

This study explores the migration of out-of-tree kernel patches in the Linux kernel across versions. Our proposed benchmark reveals that LLMs struggle with incomplete code context understanding and inaccurate migration point identification. To address these issues, we propose MIGGPT, an automated tool for migrating Linux kernel downstream patches. Our evaluation highlights MIGGPT's effectiveness and potential to advance this field.

## Acknowledgments and Disclosure of Funding

This work is partially supported by the Strategic Priority Research Program of the Chinese Academy of Sciences (Grants No.XDB0660300, XDB0660301, XDB0660302), Science and Technology Major Special Program of Jiangsu (Grant No. BG2024028), the NSF of China (Grants No. U22A2028, 62302483, 6240073476), CAS Project for Young Scientists in Basic Research (YSBR-029) and Youth Innovation Promotion Association CAS. This work is also supported by NSFC-92467102.

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

# A Benchmark

## A.1 Collection

The migration examples in our benchmark are derived from three open-source out-of-tree kernel patch projects: RT-PREEMPT [21], HAOC [16], and Raspberry Pi kernel [39]. RT-PREEMPT [4] enhances the Linux kernel's real-time performance for timing-sensitive applications like industrial control and robotics, while Raspberry Pi Linux [5] offers a lightweight kernel optimized for embedded systems. HAOC [6] improves kernel security through a "dual-kernel" design, enhancing code behavior, data access, and permission management. These projects are widely adopted in industry and open-source communities, ensuring coverage of critical out-of-tree patch scenarios. Notably, RT-PREEMPT's latest version has been integrated into the mainline Linux kernel for maintenance and no longer exists as an out-of-tree kernel patch. However, this does not impede our utilization of it for research on automated migration and maintenance of out-of-tree kernel patches.

## A.2 Examples of Benchmark

As shown in Table 1, we categorized these samples based on the difficulty of migration into two classes:

**Type 1**: This type of migration example satisfies $\Delta \neq \varnothing, \Sigma \neq \varnothing, \forall \delta \in \Delta, \forall \sigma \in \Sigma, \langle \delta, \sigma \rangle = 0$. This indicates that both the out-of-tree kernel patch and the new version of the Linux kernel have modified the code snippet, and their changes do not affect the same lines of code, meaning the modifications do not overlap or conflict with each other. As shown in Table 1 for example, $s'_{\text{old}}$ introduces additional lines of code to the function definition of `hisilicon_1980005_enable` in $s_{\text{old}}$. Conversely, $s_{\text{new}}$ both adds and removes certain lines of code within the same function definition in $s_{\text{old}}$. However, it is important to note that these modifications do not occur on the same lines of code.

**Type 2**: This type of migration example satisfies $\Delta \neq \varnothing, \Sigma \neq \varnothing, \forall \delta \in \Delta, \exists \sigma \in \Sigma, \langle \delta, \sigma \rangle \neq 0$. This indicates that both the out-of-tree kernel patch and the new version of the Linux kernel have modified the code snippet, and their changes affect the same lines of code, resulting in overlapping modifications that conflict with each other. As illustrated in Table 1, for instance, $s'_{\text{old}}$ introduces additional lines of code to the function definition of `ptep_get_and_clear` in $s_{\text{old}}$. However, $s_{\text{new}}$ refactors the same function definition into two separate function definitions, resulting in overlapping modifications that conflict with each other.

Table 10: Formalization, Counts, and Examples of the Three Types of Migration Example.

| Class | Type 1 | Type 2 |
|---|---|---|
| Formalization | $\Delta \neq \varnothing, \Sigma \neq \varnothing,$ $\forall \delta \in \Delta, \forall \sigma \in \Sigma, \langle \delta, \sigma \rangle = 0$ | $\Delta \neq \varnothing, \Sigma \neq \varnothing,$ $\forall \delta \in \Delta, \exists \sigma \in \Sigma, \langle \delta, \sigma \rangle \neq 0$ |
| Number | 80 (59.3%) | 55 (40.7%) |
| $s_{\text{old}}$ vs $s'_{\text{old}}$ |  |  |
| $s_{\text{old}}$ vs $s_{\text{new}}$ |  |  |

---

[4]https://mirrors.edge.kernel.org/pub/linux/kernel/projects/rt

[5]https://github.com/raspberrypi/linux

[6]https://gitee.com/src-openeuler/kernel/blob/master/

## B   Examples of Out-of-tree Kernel Patch Migration

As shown in Figure 7, the migration maintenance of an out-of-tree kernel patch requires integrating the modifications from the old version out-of-tree kernel patch and the modifications from the new version Linux kernel to ultimately complete the code snippet for the new version out-of-tree kernel patch.

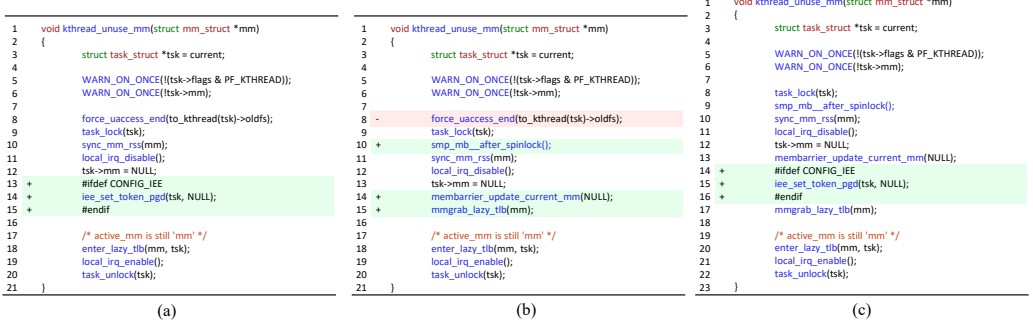

Figure 7: (a) Old version Linux kernel code snippet, with the green section indicating modifications from the old version out-of-tree kernel patch; (b) Old version Linux kernel code snippet, with the red and green sections indicating modifications for the new Linux version kernel; (c) New Linux version kernel code snippet, with the green section indicating modifications from the new version out-of-tree kernel patch.

## C   Examples of Each Challenge

### C.1   Challenge 1

```
1     static inline void __pmd_free_tlb(struct mmu_gather *tlb,
2                         pmd_t *pmdp, unsigned long addr)
3     {
4          struct ptdesc *ptdesc = virt_to_ptdesc(pmdp);
5
6          pagetable_pmd_dtor(ptdesc);
7          tlb_remove_ptdesc(tlb, ptdesc);
8     }
9     ...
10    static inline void __pte_free_tlb(struct mmu_gather *tlb,
11                        pgtable_t pte, unsigned long addr)
12    {
13         struct ptdesc *ptdesc = page_ptdesc(pte);
14
15         pagetable_pte_dtor(ptdesc);
16  +      #ifdef CONFIG_PTP
17  +          iee_tlb_remove_ptdesc(tlb, ptdesc);
18  +      #else
19             tlb_remove_ptdesc(tlb, ptdesc);
20  +      #endif
21    }
```

Figure 8: A migration case for challenge 1. The green code denotes modifications originating from the out-of-tree kernel patches.

In the migration case shown in Figure 8, we need to locate the target code snippet $s_{\text{new}}$, which defines the function `__pte_free_tlb`, within the code file file$_{\text{new}}$ of the new Linux kernel version. However, the new version file also contains a code snippet `__pmd_free_tlb` that closely resembles the target code snippet `__pte_free_tlb`. When LLMs attempt to locate the function `__pte_free_tlb` in file$_{\text{new}}$, they erroneously retrieve the similar function `__pmd_free_tlb`. This misidentification leads to errors during the migration of the out-of-tree kernel patch code. This issue highlights the challenges faced by LLMs in distinguishing between similar elements within codebases, indicating a need for improved precision in function identification and handling during the migration process.

```
 1 +    #ifdef CONFIG_IEE
 2 +    extern void set_pmd(pmd_t *pmdp, pmd_t pmd);
 3 +    #else
 4      extern pgd_t reserved_pg_dir[PTRS_PER_PGD];
 5      extern void set_swapper_pgd(pgd_t *pgdp, pgd_t pgd);
 6
 7      static inline void set_pmd(pmd_t *pmdp, pmd_t pmd)
 8      {
 9      ...
10      }
11 +    #endif
```

Figure 9: A migration case for challenge 2. In this migration case $s_{\text{old}} = s_{\text{new}}$. The green code denotes modifications originating from the out-of-tree kernel patch.

```
 1      static inline pte_t ptep_get_and_clear(struct mm_struct *mm,
 2              unsigned long addr, pte_t *ptep)
 3      {
 4          contpte_try_unfold(mm, addr, ptep, __ptep_get(ptep));
 5 +        #ifdef CONFIG_PTP
 6 +        pteval_t pteval= iee_set_xchg_relaxed(ptep, (pteval_t)0);
 7 +        pte_t ret = __pte(pteval);
 8 +        return ret;
 9 +        #else
10          return __pte(xchg_relaxed(&pte_val(*ptep), 0));
11 +        #endif
12      }
```
(a)

```
 1      static inline pte_t __ptep_get_and_clear(struct mm_struct *mm,
 2              unsigned long address, pte_t *ptep)
 3      {
 4 +        #ifdef CONFIG_PTP
 5 +        pteval_t pteval= iee_set_xchg_relaxed((pte_t *)&
 6 +                        pte_val(*ptep), (pteval_t)0);
 7 +        pte_t pte = __pte(pteval);
 8 +        #else
 9          pte_t pte = __pte(xchg_relaxed(&pte_val(*ptep), 0));
10 +        #endif
11          page_table_check_pte_clear(mm, pte);
12
13          return pte;
14      }
15
16      static inline pte_t ptep_get_and_clear(struct mm_struct *mm,
17              unsigned long addr, pte_t *ptep)
18      {
19          contpte_try_unfold(mm, addr, ptep, __ptep_get(ptep));
20          return __ptep_get_and_clear(mm, addr, ptep);
21      }
```
(b)

Figure 10: A migration case for challenge 3. (a) The legacy Linux kernel code snippet $s_{\text{old}}$. (b) The updated Linux kernel code snippet $s_{\text{new}}$. The green code denotes modifications originating from the out-of-tree kernel patch.

## C.2 Challenge 2

In the migration case shown in Figure 9, we need to locate the target code segment $s_{\text{new}}$, which encompasses lines 4 to 10, within the code file file$_{\text{new}}$ of the new Linux kernel version. However, when LLMs perform this task, they only retrieve the code segment from line 7 to line 10. As a result, the migrated custom module code exhibits deficiencies due to the missing lines. This issue underscores the limitations of LLMs in accurately identifying precise code segments, suggesting a need for enhanced alignment strategies to improve the reliability of migration tasks.

## C.3 Challenge 3

As shown in Figure 10, in the legacy Linux kernel code snippet $s_{\text{old}}$, the function `ptep_get_and_clear` is defined. In the updated Linux kernel code snippet $s_{\text{new}}$, this function has been decomposed into two separate definitions: `__ptep_get_and_clear` and `ptep_get_and_clear`. The modifications introduced by our out-of-tree kernel patch are located within the definition of the `__ptep_get_and_clear` function in the $s_{\text{new}}$ code snippet. When employing LLMs directly to retrieve $s_{\text{new}}$ from file$_{\text{new}}$, the LLMs tend to overlook the definition of `__ptep_get_and_clear`, focusing instead on the definition of `ptep_get_and_clear` present in the new version code. Consequently, during the subsequent phase of migrating the out-of-tree kernel

```
1      /* Tail call offset to jump into */
2  +   #ifdef CONFIG_HIVE
3  +   #if IS_ENABLED(CONFIG_ARM64_BTI_KERNEL)
4  +   #define PROLOGUE_OFFSET 8 + 6
5  +   #endif
6      #define PROLOGUE_OFFSET (BTI_INSNS + 2 + PAC_INSNS + 8)
7
8      static int build_prologue(struct jit_ctx *ctx, bool ebpf_from_cbpf)
9      {
10         ...
11         const struct bpf_prog *prog = ctx->prog;
12 +       #ifdef CONFIG_HIVE
13 +       const u8 base = bpf2a64[BPF_REG_BASE];
14 +       ...
15 +       #endif
16         const int idx0 = ctx->idx;
17         ...
18     }
```

Figure 11: A migration case for challenge 4. The green code denotes modifications originating from the out-of-tree kernel patch.

patch, the correct migration point cannot be identified, leading to erroneous migration. This issue highlights the difficulties LLMs face in handling the fragmentation of code during version updates, indicating a need for improved methods to accurately locate and integrate all relevant code fragments for successful migration

## C.4 Challenge 4

As shown in Figure 11, to accurately obtain the migrated out-of-tree kernel patch code $s'_{\text{new}}$, it is essential to perform two modifications on the new Linux kernel code segment $s_{\text{new}}$ (specifically, adding the code snippet #ifdef CONFIG_HIVE at two locations). However, when LLMs undertake this task, they either misidentify the migration positions or only execute one of the required modifications. This results in the failure of the out-of-tree kernel patch code migration. This issue reveals the limitations of LLMs in interpreting the precise context required for accurate migration, suggesting a need for more refined techniques to enhance the models' ability to correctly infer migration points based on the given information.

# D MIGGPT Modules

## D.1 Examples of CFP

Figure 12 illustrates a segment of code alongside its corresponding CFP. The CFP substatement in the second row of Figure 12 (b), IfdefNode, represents the second line of the code snippet in Figure 12 (a). This indicates an #ifdef statement that spans from line 2 to line 4 (pos=2, end=4) of the code segment, with the critical identifier being ARM_64_SWAPPER_USES_MAPS (name='ARM_64_SWAPPER_USES_MAPS').

## D.2 Examples of Retrieval Augmentation Module

The retrieval augmentation module is designed to address challenge 1 and challenge 3.

For challenge 1, we construct a "Structure Prompt" to specify the signatures of the code snippet $s_{\text{old}}$. By constructing the code fingerprint structure $\text{CFP}_{\text{old}}$ from $s_{\text{old}}$ as shown in Figure 8, we can extract FuncDef statements that contain the code signatures (Figure 13), thereby generating a "Structure Prompt" that describes these signatures. Consequently, the LLM will focus its attention on the function definition __pte_free_tlb rather than on the similar function definition __pmd_free_tlb. This Structure prompt enhances the LLM's ability by providing a precise description of the target code, allowing the LLM to focus more accurately on the relevant code snippet and improving the precision of the retrieval.

For challenge 3, we extract the associated function calls of the code snippet to provide comprehensive code context. As shown in Figure 10 (b), when retrieving $s_{\text{new}}$, the LLM can only find the definition snippet of the function ptep_get_and_clear (lines 16-21) and overlooks the definition snippet of the internally called function __ptep_get_and_clear (lines 1 to 14). To address this challenge,

```
1      /* We intend to enable IRQs */
2      #ifdef ARM_64_SWAPPER_USES_MAPS
3      #include <iee/setup.h>
4      #endif
5
6      static inline void local_daif_mask(int set_mm)
7      {
8          ...
9          asm volatile(
10             "msr daifset, #0xf"
11             :
12             :
13             : "memory");
14         ...
15         /* Don't really care for a dsb here */
16           if (system_uses_nmi())
17              _allint_set();
18         trace_hardirqs_off();
19     }
```

(a)

```
1  CommentNode(pos=1, end=1, content='We intend to enable IRQs ')
2  IfdefNode(pos=2, end=4, name='ARM_64_SWAPPER_USES_MAPS')
3  IncludeNode(pos=3, end=3, content='iee/setup.h')
4  FuncDef(pos=6, end=19, name='local_daif_mask',
5              type=['static', 'inline', 'void' ],
6              param=[VarDec(name='set_mm', type=['int'])])
7  ...
8  ASMNode(pos=9, end=13,
9              context='\"msr daifset, #0xf\"\n:\n:\n:\"memory\"')
10 ...
11 CommentNode(pos=15, end=15,
12              content='Don't really care for a dsb here ')
13 IfNode(pos=16, end=17, inline_fuccalls=[
14              FuncCall(pos=17, end=17, name='system_uses_nmi')])
15 FuncCall(pos=17, end=17, name='_allint_set')
16 FuncCall(pos=18, end=18, name='trace_hardirqs_off')
```

(b)

Figure 12: (a) A code snippet. (b) Corresponding CFP of the code snippet.

```
1  FuncDef(name='__pte_free_tlb', type=['static inline'], param=[
2              VarDec(name='tlb', type=['struct', 'mmu_gather', '*']),
3              VarDec(name='pte', type=['pgtable_t']),
4              VarDec(name='addr', type=['unsigned', 'long'])
5              ])
```

Figure 13: The CFP statement on line 10 of Figure 8

it is necessary to supplement the initially retrieved $s_{tmp}$ from file$_{new}$ with its invoked associated functions, ultimately obtaining a complete code snippet $s_{new}$. It should be noted that the function `ptep_get_and_clear` often invokes many functions (such as `contpte_try_unfold` on line 19), which also appear in $s_{old}$ (line 4 of Figure 10 (a)) and are not what we require. Therefore, we need to select only those associated functions that are invoked within $s_{tmp}$ but not by $s_{old}$ to form the complete code snippet $s_{new}$.

## D.3  Examples of Migration Augmentation Module

The migration augmentation module is primarily designed to address challenge 4. Specifically, as shown in Figure 11, we conduct a comparative analysis between the code fingerprint structures CFP$_{old}$ and CFP$'_{old}$ of the code snippets to ascertain that there are two primary migration points. The first point is located after the comment statement `Tial call offset...` and before the macro definition statement `#define PROLOGUE_OFFSET...`. The second point is situated after the statement `const struct bpf_prog...` and before the statement `const int idx0=ctx->idx`. By constructing the "Location Prompt", we enable the LLM to precisely locate the migration points, thereby successfully completing the task of migrating and maintaining the out-of-tree kernel patch.

## D.4  Algorithm and Prompts

Algorithm 1 and Algorithm 2 respectively illustrate the specific details of target code snippet retrieval and code migration processes. We also present all the prompts utilized by MIGGPT. As shown in Figure 14, when retrieving the target code snippet $s_{new}$, we construct

---

**Algorithm 1** Retrieval of the target code snippet $s_{\text{new}}$

---

1: **Input:** $(s_{\text{old}}, \text{file}_{\text{new}})$, LLM, and maximum query count $m$
2: **Output:** $s_{\text{new}}$
3: Generating $\text{CFP}_{\text{old}}$ form $s_{\text{old}}$
4: Preparing $RetrievalTaskPrompt$
5: Preparing $ExpertPersonaPrompt$
6: $\mathcal{S} \leftarrow \text{Extractsignature}(\text{CFP}_{\text{old}})$
7: $StructurePrompt \leftarrow \text{Prompt}(\mathcal{S})$
8: $\mathcal{A} \leftarrow \text{Extractanchor}(\text{CFP}_{\text{old}})$
9: $AlignmentPrompt \leftarrow \text{Prompt}(\mathcal{A})$
10: $RetrievalPrompt \leftarrow$
11:     $RetrievalTaskPrompt + StructurePrompt$
12:     $+AlignmentPrompt + ExpertPersonaPrompt$
13: **while** $q <= m$ **do**
14:     $s_{tmp} \leftarrow \text{LLM}(RetrievalPrompt, s_{\text{old}}, \text{file}_{\text{new}})$
15:     Generating $\text{CFP}_{\text{tmp}}$ from $s_{tmp}$
16:     **if** $\text{find}(\mathcal{S}, s_{tmp})$ **then**
17:         **break**
18:     **end if**
19:     $q \leftarrow q + 1$
20: **end while**
21: $\mathcal{F}_{\text{old}} \leftarrow \text{Funccall}(\text{CFP}_{\text{old}})$
22: $\mathcal{F}_{\text{tmp}} \leftarrow \text{Funccall}(\text{CFP}_{\text{tmp}})$
23: $s_{\text{new}} \leftarrow s_{\text{tmp}} + \text{FindCode}(\mathcal{F}_{\text{tmp}} \setminus \mathcal{F}_{\text{new}}, \text{file}_{\text{new}})$
24: **return** $s_{\text{new}}$

---

**Algorithm 2** Migration of code snippet $s'_{\text{new}}$

---

1: **Input:** $(s_{\text{old}}, s'_{\text{old}}, s_{\text{new}})$ and LLM
2: **Output:** $s'_{\text{new}}$
3: Generating $\text{CFP}_{\text{old}}, \text{CFP}'_{\text{old}}$ form $s_{\text{old}}$ and $s'_{\text{old}}$
4: Preparing $MigrationTaskPrompt$
5: Preparing $ExpertPersonaPrompt$
6: $\mathcal{P} \leftarrow \text{PinpointMigrationLocation}(\text{CFP}_{\text{old}}, \text{CFP}'_{\text{old}})$
7: $LocationPrompt \leftarrow \text{Prompt}(\mathcal{P})$
8: $MigrationPrompt \leftarrow MigrationTaskPrompt$
9:     $+LocationPrompt + ExpertPersonaPrompt$
10: $s'_{\text{new}} \leftarrow \text{LLM}(LocationPrompt, s_{\text{old}}, s'_{\text{old}}, s_{\text{new}})$
11: **return** $s'_{\text{new}}$

---

the $Retrieval\ Prompt$ to query LLMs. Specifically, we employ $Task\ Prompt$ 1 to describe the task and $Expert\ Persona\ Prompt$ to standardize the output format of LLMs. Additionally, $StructurePrompt$ and $AlignmentPrompt$ are used to enhance the retrieval capabilities of the LLMs. When generating the migrated code snippet $s'_{\text{new}}$, we construct the $Migration\ Prompt$ to query LLMs. Specifically, we utilize $Task\ Prompt$ 2 to describe the task and $Expert\ Persona\ Prompt$ to standardize the output format of the large language model. Additionally, $LocationPrompt$ is employed to enhance the migration capabilities of the LLM.

## D.5 CFP

Algorithm 3 illustrates the specific details about generating CFP. The code snippet $s$ is tokenized, and nested scopes (e.g., `{`,`}`, `#ifdef`/`#endif`) are identified via bracket matching. Critical symbols (e.g., `{`, `}`, `#ifdef`, `func()`) demarcate code blocks. Function calls (e.g., `foo()`) are detected through pattern matching on token sequences (e.g., identifier followed by `()`). Associated functions are identified by analyzing call statements within code blocks, avoiding call graph construction. An example of step-by-step CFP generation is illustrated in Figure 16.

**Algorithm 3** The generation of CFP

1: **Input:** code snippet $s$
2: **Output:** $\text{CFP}_s$
3: Initialize list $\text{CFP}_s$
4: **for** $line$ in $s$ **do**
5:     $Node \leftarrow \text{IdentifyType}(line)$
6:     $(Node.pos, Node.end) \leftarrow \text{IdentifyScope}(line)$
7:     **if** $Node \in \{\text{FuncDef}\}$ **then**
8:         $Node.parameter \leftarrow \text{CFP}(\text{InternalStatement}(Node))$
9:     **end if**
10:     **if** $Node \in \{\text{if}, \text{else}, \text{while}, \text{for}, \text{do}, \text{switch}\}$ **then**
11:         $Node.inline\_fuccall \leftarrow \text{Funcall}(\text{InternalStatement}(Node))$
12:     **end if**
13:     $Node.content \leftarrow \text{Content}(Node)$
14: **end for**
15: $\text{CFP}_s \leftarrow \text{CFP}_s \cup \{Node\}$
16: **return** $\text{CFP}_s$

<table>
<tr><td>

**Retrieval Prompt**

**Retrieval Task Prompt**: We are facing a challenge that requires your specialized knowledge and expertise. We need to locate a corresponding segment of code, indicated as `part_new`, within a C file named `new.c` that matches semantically with a provided code snippet labeled as `part_old`. Given that `part_new`, the target code segment, originates from modifications made to `part_old`, it is essential to identify this correspondence accurately. The starting point for your task involves comparing the following `part_old`: {code of $v_{old}$}. And the entire context available in the `new.c`: {code of file$_{new}$}.

**Structure Prompt**: It appears that `part_old` encompasses the definition of the function `{target function signature of $\text{CFP}_{old}$}`. Your role is to pinpoint the matching code segment `part_new` within `new.c`. Please ensure that the identified function definitions are solely derived from `new.c`. Avoid constructing false code snippets by using the function definitions from `part_old`.

**Alignment Prompt**: To facilitate the search, you may need to align `part_new` using the initial line `{head anchor statement of $\text{CFP}_{old}$}` and the final line `{tail anchor statement of $\text{CFP}_{old}$}` from `part_old`.

**Expert Persona Prompt**: You are an expert in Linux Kernel development and coding. We kindly ask you to respond with a Markdown-formatted string within a code block that starts and ends with triple backticks (```). The response should strictly contain the identified `part_new` without providing additional analysis or using a list to store lines of code.

</td><td>

**Migration Prompt**

**Migration Task Prompt**: I am reaching out to you with a specialized code migration task where your expertise in Linux kernel development would be invaluable. Your assistance will help ensure the successful adaptation of existing code to the latest version of the Linux kernel. For this task, I will provide three code snippets for your consideration. Code Snippet 1: The old version of the Linux kernel code snippet, which we will refer to as `part_old`: {code of $v_{old}$}. Code Snippet 2: The corresponding code developed based on the old version of the Linux kernel code snippet `part_old`, referred to as `part_old_patched`: {code of $v'_{old}$}. Code Snippet 3: The new version of the Linux kernel code snippet, denoted as `part_new`: {code of $v_{new}$}.

**Location Prompt**: Upon preliminary analysis, it appears that there is {number of modifications} specific area within `part_old_patched` that requires modification: The first modification should be made situated after the line containing {head location statement of $\text{CFP}_{old}$}, and before the line containing {tail location statement of $\text{CFP}_{old}$} with the change being {analysis of $\text{CFP}_{old}$ and $\text{CFP}'_{old}$}...... It's likely that similar adjustments will need to be made within `part_new` to maintain functionality and compatibility. Given your extensive knowledge and experience in this field, could you kindly assist by generating the corresponding code snippet `part_new_patched` developed on `part_new`?

**Expert Persona Prompt**: You are an expert in Linux Kernel development and coding. We kindly ask you to respond with a Markdown-formatted string within a code block that starts and ends with triple backticks (```). The response should strictly contain the identified `part_new` without providing additional analysis or using a list to store lines of code.

</td></tr>
</table>

Figure 14: The prompts of MIGGPT.

# E  More Results of MIGGPT

We have also tested the performance of MigGPT on DeepSeek-V2.5 [7], as shown in Table 11 and Table 12, MIGGPT outperforms the vanilla LLM.

Table 11: The accuracy of the MIGGPT-augmented LLMs compared to vanilla LLMs on retrieving target code snippets.

| LLM | Method | Type 1 (80) | | | Type 2 (55) | | | All (135) | | | Average Query Times |
|---|---|---|---|---|---|---|---|---|---|---|---|
| | | Best Match | Semantic Match | Human Match | Best Match | Semantic Match | Human Match | Best Match | Semantic Match | Human Match | |
| DeepSeek-V2.5 | Vanilla | 61.25% | 66.25% | 62.50% | 65.45% | 69.09% | 67.27% | 62.96% | 67.40% | 64.44% | 1.00 |
| | MIGGPT | 95.00% | 97.5% | 96.25% | 87.27% | 90.90% | 90.90% | 91.85% | 94.81% | 94.07% | 1.22 |

# F  Settings

## F.1  Platform

Our experiments were conducted on a system equipped with an AMD Ryzen 9 7900X 12-core CPU, 32GB of RAM, running on Ubuntu 22.04.3 LTS.

## F.2  Contextual Information

The contextual information given to the LLM is identical for both the "two-step strategy" (baseline vanilla approach in Tables 1 and 2) and the "one-step strategy", so the comparison is fair (details are shown in Table 13). The contextual information includes the old version code $s_{old}$, $s'_{old}$, and the new

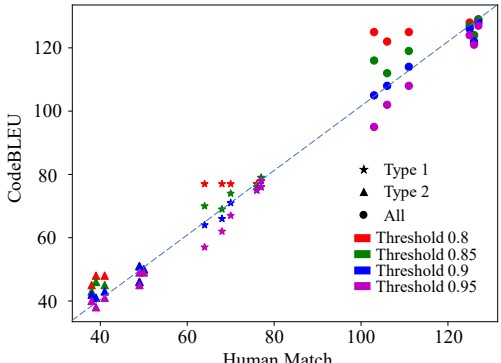

Figure 15: Comparison of Consistency with Human Match at Different Thresholds for CodeBLEU.

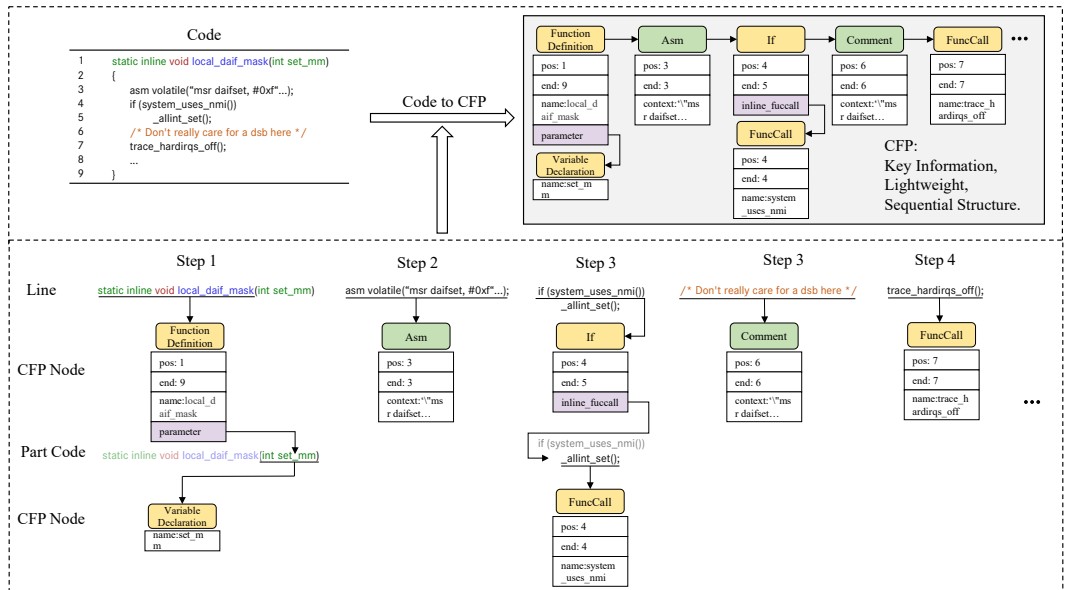

Figure 16: An example of step-by-step CFP generation.

version file `new.c`. As shown in Figure 5 of the paper, the "two-step strategy" is clearly superior to the "one-step strategy".

We provided the key migration information extracted by CFP and the code information ($s_{old}$, $s\prime_{old}$, `new.c`) as context to the "one-step strategy" (One-step + CFP) and compared it with MigGPT (Two-step + CFP). As shown in Table 14, formulating the migration task as a two-step process demonstrates performance advantages.

## F.3   The Reliability of Human Match

We strictly adhere to defined steps and principles for Human Match testing, ensuring reliable post-migration code functionality verification:

- Cross-validation: 5 experienced Linux kernel engineers independently validate results.

- Triple-blind voting: Each example is evaluated by 3 randomly assigned engineers.

- Criteria: (1) syntactic consistency, which ensures the preservation of original patch logic through structural adaptations (e.g., variable renaming while maintaining control flow); and (2) semantic correctness, verifying functional equivalence between migrated code and human-generated ground truth patches.

Table 12: The accuracy of the MIGGPT-augmented LLMs compared to vanilla LLMs on the migration task of target code snippets.

| LLM | Method | Type 1 (80) | | | Type 2 (55) | | | All (135) | | |
|-----|--------|------------|----------------|-------------|------------|----------------|-------------|------------|----------------|-------------|
| | | Best Match | Semantic Match | Human Match | Best Match | Semantic Match | Human Match | Best Match | Semantic Match | Human Match |
| DeepSeek-V2.5 | Vanilla | 11.25% | 16.25% | 18.75% | 9.09% | 27.27% | 21.82% | 10.37% | 20.74% | 20.00% |
| | MIGGPT | 67.50% | 80.00% | 80.00% | 56.36% | 74.55% | 70.91% | 62.96% | 77.78% | 76.30% |

Table 13: The contextual information of various methods.

| Context | One-step strategy | Two-step strategy (Vanilla) | One-step strategy + CFP | MigGPT (Two-step strategy + CFP) |
|---------|-------------------|-----------------------------|-------------------------|----------------------------------|
| Code Information | ✓ | ✓ | ✓ | ✓ |
| CFP Information | ✗ | ✗ | ✓ | ✓ |

Adherence to these provisions enhances the reliability and credibility of our Human Match metric, ensuring rigorous alignment with established evaluation standards.

## F.4 Compilation Success Rate

We conducted additional compilation success rate experiments on migrated patches. For each migration sample, we replace the generated code with MigGPT into the patched version of the new kernel in a containerized environment and attempt to compile the modified kernel. The compilation success rate is the ratio of successfully compiled samples to all types of migration examples.

## F.5 Line Edit Distance

The line edit distance is a measure of the difference between two code snippets. It is defined as the minimum number of single-line edit operations (insertions, deletions, or substitutions) required to transform one line into another.

Given two code snippets $A = \{a_i\}_{i=1}^n$ and $B = \{b_j\}_{j=1}^m$ with line lengths $|A| = n$ and $|B| = m$, the line edit distance $D(A, B)$ can be defined recursively as follows:

$$
D(A, B) = \begin{cases} \max(n, m) & \text{if } \min(n, m) = 0, \\ \min \begin{cases} D(\text{prefix}(A, n-1), B) + 1, \\ D(A, \text{prefix}(B, m-1)) + 1, \\ D(\text{prefix}(A, n-1), \text{prefix}(B, m-1)) + \mathbb{I}(a_n \neq b_m) \end{cases} & \text{otherwise.} \end{cases}
\tag{1}
$$

Where:

1. $\text{prefix}(A, k) = \{a_i\}_{i=1}^k$ denotes the first $k$ lines of code snippet $A$.

2. $\mathbb{I}(a_i \neq b_j)$ is an indicator function that equals 1 if $a_i \neq b_j$ and 0 otherwise.

3. The three cases in the recursion correspond to:
   1) Deletion: Delete the last line of $A$ and compute $D(\text{prefix}(A, n-1), B)$.
   2) Insertion: Insert the last line of $B$ into $A$ and compute $D(A, \text{prefix}(B, m-1))$.
   3) Substitution: Replace the last line of $A$ with the last line of $B$ (if they differ) and compute $D(\text{prefix}(A, n-1), \text{prefix}(B, m-1))$.

## F.6 Threshold of CodeBLEU

CodeBLEU [40] is an automated metric designed to evaluate the quality of code generation, specifically tailored for tasks involving the generation of programming code. By integrating both syntactic and semantic features of code, CodeBLEU provides a similarity score ($[0, 1]$) between two code snippets. We employ CodeBLEU as a measure of "semantic match" and investigate the alignment between CodeBLEU-based "semantic matches" and "human matches" across various thresholds. As illustrated in Figure 15 and Table 15, we identify a threshold of 0.9 as optimal for our proposed benchmark, ensuring a high degree of consistency between "semantic matches" derived from CodeBLEU and those determined by human evaluation.

Table 14: The accuracy of MIGGPT and "One-step + CFP" on the migration task.

| Method | Best Match | Semantic Match |
|---|---|---|
| One-step + CFP | 38.52% | 42.96% |
| MIGGPT | 62.96% | 80.00% |

Table 15: The results of MIGGPT, compared to the ground truth, are presented in terms of the number of correct examples under both CodeBLEU "semantic match" and "human match". Here, "CodeBLEU-0.8" denotes a CodeBLEU classification threshold set at 0.8.

| Metric | Type | GPT-4-turbo | | DeepSeek-V2.5 | | DeepSeek-V3 | | Average | |
|---|---|---|---|---|---|---|---|---|---|
| | | Retrieval | Migration | Retrieval | Migration | Retrieval | Migration | Retrieval | Migration |
| Human Match | Type1 | 77 | 68 | 77 | 64 | 76 | 70 | 77 | 67 |
| | Type2 | 49 | 38 | 50 | 39 | 49 | 41 | 49 | 39 |
| | All | 126 | 106 | 127 | 103 | 125 | 111 | 126 | 107 |
| CodeBLEU-0.8 | Type1 | 78 | 77 | 79 | 77 | 77 | 77 | 78 | 77 |
| | Type2 | 46 | 45 | 50 | 48 | 51 | 48 | 49 | 47 |
| | All | 124 | 122 | 129 | 125 | 128 | 125 | 127 | 124 |
| CodeBLEU-0.85 | Type1 | 78 | 69 | 79 | 70 | 76 | 74 | 78 | 71 |
| | Type2 | 46 | 43 | 50 | 46 | 51 | 45 | 49 | 45 |
| | All | 124 | 112 | 129 | 116 | 127 | 119 | 127 | 116 |
| CodeBLEU-0.9 | Type1 | 76 | 66 | 78 | 64 | 75 | 71 | 76 | 67 |
| | Type2 | 46 | 42 | 50 | 41 | 51 | 43 | 49 | 42 |
| | All | 122 | 108 | 128 | 105 | 126 | 114 | 125 | 109 |
| CodeBLEU-0.95 | Type1 | 76 | 62 | 78 | 57 | 75 | 67 | 76 | 62 |
| | Type2 | 45 | 40 | 49 | 38 | 49 | 41 | 48 | 40 |
| | All | 121 | 102 | 127 | 95 | 124 | 108 | 124 | 102 |

## F.7 Variant of MIGGPT

We implement four variants for the ablation study:

1. MigGPT-No-Retrieval-Augmentation: The Retrieval Augmentation Module is deactivated, causing no constraint on the structure of code snippets.

2. MigGPT-No-Retrieval-Alignment: The Retrieval Alignment Module is deactivated, leading to the absence of descriptions for the starting and ending line information of code snippets.

3. MigGPT-No-Migration-Augmentation: The Migration Augmentation Module is disabled. The LLMs will not have the assistance of additional analytical information when completing migration tasks.

4. MigGPT-No-CFP: Replace all components of MIGGPT that require CFP participation (including code snippet invocation relationship analysis, anchor function identification, and migration location detection) with implementations utilizing LLMs.

# G  Discussion

## G.1  Type 1 Migration Sample

Traditional methods like FixMorph [43], SyDIT [31], TSBPORT [53], and PPatHF [37] target single-step migration and cannot solve the target code retrieval problem, while MIGGPT tackles both problems, which is harder and more practical. Besides, traditional methods struggle with Type 1 cases due to their reliance on static alignment and predefined transformation rules. As an example shown in Figure 17, when backporting a patch that modifies `compute_stats()` in the old kernel, FixMorph relies on static alignment (e.g., matching function names like `process_data`). However, since `compute_stats()` is now a standalone function in the new kernel, traditional methods like FixMorph cannot generate transformation rules for the split code structure, as its AST differencing assumes code blocks stay within the same function. This illustrates how traditional methods struggle with Type 1's non-conflicting but structurally divergent changes.

## G.2  CFP and Intermediate Representations

A pertinent research question emerges: Could the structural semantics captured through code snippet intermediate representations (IR) provide enhanced contextual signals for guiding LLM-based

```
1   void process_data() {
2       validate_input();
3       compute_stats();        // Patch modifies this line
4       log_results();
5   }
```

```
1   void compute_stats() { ... }
2   void process_data() {
3       validate_input();
4       compute_stats();        // Unmodified line
5       log_results();
6   }
```

(a)                                                    (b)

Figure 17: (a) A code snippet of old kernel. (b) The code snippet is refactored into modular functions in the new kernel.

code migration processes? The answer is negative. While IR-based approaches excel at syntax normalization, our CFP method prioritizes two critical requirements for kernel patch migration:

- Context Preservation: Kernel patches often contain conditional compilation macros, inline assembly, and annotations stripped during IR generation (e.g., preprocessing eliminates macros). CFP retains these alongside code structure (function signatures, control-flow anchors) to provide LLMs with a full migration context.
- Non-Compilable Code Support: Experimental/unmerged patches (e.g., ARM64-specific optimizations) may fail compilation, making IR extraction impossible. CFP operates directly on source fragments, even for"broken" code in development.

This demonstrates the necessity of employing CFP in MIGGPT.

## G.3    Generalization of MIGGPT

MIGGPT demonstrates innovative advancements through its dual strengths of generalizability and domain-specific optimization. While initially designed for out-of-tree kernel patches, its core methodology addresses universal challenges in LLM-based code migration, such as resolving structure conflicts, precisely aligning code boundaries, reconstructing missing context, and establishing migration localization mechanisms, forming a framework extensible to multiple domains (CVE patch backporting [43, 53], forked code porting [37, 55]). The architecture incorporates its Code Fingerprint structure, which embeds Linux kernel-specific optimizations through preserving inline assembly instructions and kernel macro patterns, interpreting kernel-specific comment conventions, and adapting to coding norms like function chaining. This technical design achieves a balance between cross-domain adaptability and deep specialization, with the CFP serving as a modular component that enhances Linux kernel patch migration while maintaining the system's capacity for expansion into other technical ecosystems. The solution thus enables bidirectional scalability, supporting both horizontal domain transfer and vertical technical refinement.

## G.4    Structured Analysis of CFP

The high-level idea of using program analysis techniques to create structured representations of code to guide LLMs is an emerging area of research. We have compared MigGPT with existing research in this emerging area. For instance, works such as AutoOS [3] and BYOS [29] utilize heuristic tree structures to assist LLMs in optimizing kernel configurations during the deployment of operating systems. Similarly, LLM-based fuzzing tools [49, 18] employ formal templates to guide LLMs in generating more effective test cases. These approaches, much like MigGPT, leverage deterministic information such as trees, graphs, and lists to mitigate the inherent uncertainty and randomness of LLMs, thereby improving performance across various application tasks. However, it's worth noting that the specific deterministic information used and its data structure are often tailored to the particular task at hand.

## G.5    Deprecated System Calls

Once a system call is added to the kernel, it's generally supported indefinitely to avoid breaking existing software that relies on it. However, kernels do evolve, introducing new system calls that offer more robust, secure, or efficient functionalities. These new features can, in some cases, functionally supersede older, more limited mechanisms. When a patch migration involves new system calls

introduced in a newer kernel version, MigGPT is well-equipped to handle such scenarios. In these cases, the new version target code ($s_{new}$) will contain examples of how the new system call is used. MigGPT can then refer to these usage examples in $s_{new}$ to complete the migration and generate a corresponding patch that aligns with the new API. Certainly, in this scenario, migration types can still be categorized according to the rules in Table 1. If the new kernel version and patch modifications don't overlap in location, it's Type 1. Otherwise, it's Type 2.

## G.6 Impact Statement

This work advances the field of automated software maintenance by introducing MIGGPT, a framework that leverages LLMs to automate the migration and maintenance of out-of-tree Linux kernel patches. By reducing the manual effort and costs associated with these tasks, our research has the potential to improve the efficiency and reliability of software systems, benefiting industries that rely on stable and up-to-date infrastructure.

However, the adoption of such automation tools also raises ethical considerations. For example, automating tasks traditionally performed by specialized engineers may impact job roles, necessitating workforce adaptation. Additionally, the reliance on LLMs for critical maintenance tasks requires rigorous validation to ensure accuracy and avoid potential risks to system stability and security.

While our primary focus is on technical advancements, we acknowledge the broader societal implications of automating complex engineering processes. This study lays the foundation for future research and encourages ongoing discussions on the responsible use of AI in software maintenance, balancing innovation with ethical considerations.

