# OpenReview forum: "MigGPT: Harnessing Large Language Models for Automated Migration of Out-of-Tree Linux Kernel Patches Across Versions"
_NeurIPS.cc/2025/Conference — NeurIPS 2025 spotlight_

### Official Review · Reviewer_NY3E · 2025-07-01

**Clarity:** 4
**Significance:** 4
**Originality:** 3
**Rating:** 4
**Confidence:** 4

**Summary:**

The paper proposes a MIGGPT, a novel framework that employs code fingerprints (CFPs) to enhance Large Language Models in backporting out-of-tree Linux Kernel patches. The paper identifies key challenges in using LLMs for this task, such as namespace interference and the absence of key code information. To address these, MIGGPT leverages a novel data structure CFP to augment LLM's capability in backporting Linux Kernel patches across different versions. The experimental results demonstrate that the proposed MIGGPT framework outperforms the direct application of vanilla LLMs, achieving more than a 70% improvement in the completion rate.

**Questions:**

I appreciate the authors' efforts in applying LLMs to backport out-of-tree Linux Kernel patches. The proposed method provides a novel way to constrain LLMs to focus on the specific context of the program, which effectively prevents the potential issues of directly using LLMs for program-repair tasks, such as hallucination. While the experimental results show MIGGPT works excellently on the task, I am still concerned with a few aspects of the proposed method.

### On the code fingerprint (CFP)

While the paper claims that using Abstract Syntax Trees (ASTs) is unable to tackle some scenarios during the analysis since it is highly tight to the compilation process, I was wondering if the tools commonly used in static analysis such as the Intermediate Representation (IR) of the kernel program could be helpful to distinguish the patches from similar contexts. Since IR (either LLVM IR or Gimple) is specifically designed to address the syntax sugars, would that be a better choice for this task?

### On the wider application/generalization of MIGGPT

The proposed framework significantly improves the capability of LLMs to backport out-of-tree patches in Linux Kernel. However, the proposed framework gives me a feeling that it is quite a general method of backporting patches from the mainstream project to the forks or distributions. Is it possible to extend the pack backporting approach to other similar scenarios? If so, could the authors generally discuss a wider application of MIGGPT? I would like to suggest adding a minor discussion section in the paper, which could highlight the future work.

**Ethical Concerns:**

["NO or VERY MINOR ethics concerns only"]

**Final Justification:**

I have went through authors' response and other reviews. My concerns have been addresses. I would like to keep my decision and recommend the acceptance of this paper.

**Limitations:**

Yes.

**Paper Formatting Concerns:**

N/A.

**Quality:**

3

**Strengths And Weaknesses:**

### Strengthens

- The task is important. LLMs are applied in a reasonable way to reduce human labor.
- The experiments are well designed, including the ablation experiment.
- The experimental results are impressive.

### Weaknesses

- The proposed LLM-based approach seems not to be limited to the Linux Kernel vulnerability patching only. I would like to recommend that the authors discuss the potential generalization of MIGGPT to a wider application.

-  Potential runtime overhead brought by using LLMs.

See more details in `Questions`.

---

> ### Author Rebuttal · Authors · 2025-07-30
>
> Thank you for your review comments. Your concerns regarding the discussion on MigGPT's generalization, IR-based approaches, and runtime overhead have all been addressed. We discussed and supplemented the experiment to address the concern. Please refer to the specific responses below and our rebuttal for more information.
>
> ---
>
> **Weakness 1.** Generalization of MigGPT
>
> We appreciate you bringing up the crucial question of MigGPT's generalizability to other migration scenarios and domains. In response, we've incorporated new experimental findings:
>
> - **Demonstrated Generalizability to Other Migration Tasks:** As explicitly detailed in Table 5 of our paper, MigGPT was successfully deployed in a CVE patch backporting task—specifically, migrating a CVE patch from its discovery version to an earlier iteration. In this task, we only need to modify MigGPT's task prompt (to describe the details of the CVE patch backporting task); no modifications are required for the CFP. The results show that MigGPT consistently outperformed existing solutions.
> - **Enhanced Cross-Language Generalizability:** We randomly selected 31 migration examples from our benchmark and translated them into Python. With only minimal modifications required for the statement tokenization and statement element extraction functions of CFP to accommodate Python syntax, MigGPT efficiently processed these examples. The results show an average semantic match performance improvement of 16.13% when compared to the vanilla approach, thereby underscoring MigGPT's robust effectiveness beyond its original C-language application.
>
> | LLM | Method | Best Match | Semantic Match |
> | --- | --- | --- | --- |
> | GPT-4-turbo | Vanilla | 61.29% | 70.97% |
> | GPT-4-turbo | MigGPT | 80.65% | 87.10% |
> | DeepSeek-V3 | Vanilla | 64.52 | 74.19% |
> | DeepSeek-V3 | MigGPT | 83.87% | 90.32% |
>
> ---
>
> **Weakness 2.** Runtime Overhead
>
> We would like to thank the reviewer for the insightful comment.  As shown in **Table 6** of our manuscript, we conducted a comparison between MigGPT and human experts in terms of time cost for performing code migration tasks.  The results demonstrate that MigGPT required only **2.08%** of the average time taken by human experts.  This highlights the capability of MigGPT to automate the code migration process, significantly reducing human effort and achieving substantial efficiency improvements.
>
> ---
>
> **Question 1.** CFP and Intermediate Representations
>
> Thank you for the suggestion. While IR-based methods are effective at normalizing syntax, CFP is designed to meet two essential needs in kernel patch migration:
>
> - **Context Preservation**: Kernel patches often include macros, inline assembly, and annotations that are lost during IR generation. CFP retains this information, along with structural elements like function signatures and control-flow anchors, to give the LLM a richer migration context.
> - **Support for Non-Compilable Code**: Early-stage or experimental patches may not compile, making IR extraction infeasible. CFP works directly on source code fragments, even if they are incomplete or broken.
>
> ---
>
> **Question 2.** Generalization of MigGPT
>
> Please refer to our response to Weakness 1

---

> > ### Comment · Reviewer_NY3E · 2025-08-01
> >
> > Thanks for your rebuttal comments. While your answer has addressed all my major concerns, I still would like to keep my original decision. I would be happy to see the acceptance of this paper.

---

> > > ### Author Response · Authors · 2025-08-01
> > >
> > > Thank you very much for your thoughtful feedback and for reconsidering our rebuttal. We sincerely appreciate your time and effort in evaluating our work. We are glad to hear that our responses have addressed your major concerns, and we truly value your positive recommendation.

---

### Official Review · Reviewer_Nf59 · 2025-07-03

**Clarity:** 3
**Significance:** 4
**Originality:** 3
**Rating:** 5
**Confidence:** 4

**Summary:**

The authors propose an LLM-based framework targeting the automation of maintaining out-of-tree Linux Kernel Patches across different kernel versions. To do so, they leverage what they define as Code Fingerprints, a sequential structure that aims to preserve code context for code segments and to facilitate migration. Further, they design their framework in a modular way to address challenges that they identify. To address structural ambiguity (similar function signature) and missing associated fragments (for example, due to refactoring work), they design a retrieval augmentation module. To address boundary ambiguity, i.e. patch anchor points can change across Linux versions, they design a Retrieval Alignment module. Finally, they built their Migration Augmentation module with the potential of ambiguous application points in mind to address a final challenge they identified. To evaluate their proposed framework on their proposed problem domain, the authors devise a new benchmark using RT-PREEMPT, RPi Linux, and HAOC. They focus on two types of migrations: those whose updated line-spans intersect, and those that do not. The latter form a more difficult category as this can signal refactoring, or modularisation, etc. This new benchmark focuses on the maintenance use case, in contrast to previous work, which focused on (security) back-porting. They evaluate their framework on two benchmarks, the one proposed and FixMorph's CVE patch back-porting framework. MigGPT shows consistent improvement on Best Match, Semantic Match, and Human Match across all considered LLMs. Further, it also shows significant improvement in code compilation on GPT-4-turbo and DeepSeek-V3.

**Questions:**

Q1. In your dataset, has it been the case that the out-of-tree patch-set used _now_ deprecated system calls? What would the expected LLM output be in such a case? I feel there can exist a Type 3 (perhaps in other systems if the LK does not show such cases) where the API/system calls have evolved and the patch itself should be augmented, not just determining where to apply it unaltered.

- Side remark: While I appreciate the argument for code fingerprints in §5.3/G.2, the argument is concerning the AST of the full language grammar. Alternatives could be considered, specifically island grammars. Interesting structures could be relegated to "land" and parsed, while noise could be relegated to "water". This would not address the AST depth issue; however, it can address the issue of focusing on specific patterns as well as preserving comments or other information usually not included in traditional ASTs (see H-AST in [+3])
[+1]Balland, E., Kirchner, C. and Moreau, P.E., 2006, July. Formal islands. In International Conference on Algebraic Methodology and Software Technology (pp. 51-65). Berlin, Heidelberg: Springer Berlin Heidelberg.
[+2]Afroozeh, A., Bach, J.C., Van den Brand, M., Johnstone, A., Manders, M., Moreau, P.E. and Scott, E., 2013. Island grammar-based parsing using gll and tom. In Software Language Engineering: 5th International Conference, SLE 2012, Dresden, Germany, September 26-28, 2012, Revised Selected Papers 5 (pp. 224-243). Springer Berlin Heidelberg.
[+3] L. Ponzanelli, A. Mocci and M. Lanza, "StORMeD: Stack Overflow Ready Made Data," 2015 IEEE/ACM 12th Working Conference on Mining Software Repositories, Florence, Italy, 2015, pp. 474-477, doi: 10.1109/MSR.2015.67.

**Ethical Concerns:**

["NO or VERY MINOR ethics concerns only"]

**Final Justification:**

After reading the author's responses and other reviews, I maintain my positive score.

**Limitations:**

Limitations are reasonably addressed.

**Paper Formatting Concerns:**

N/A, paper seems to follow formatting guidelines.

**Quality:**

4

**Strengths And Weaknesses:**

+ S1 Strong results on patch migration, especially the line edits result in §6.4, suggesting that the explored patch types are reasonably automatable.
+ S2 Detailed analysis of the expected data shape (out-of-tree diffs and Linux kernel version update diffs), justifying the system design from the data.
+ S/W(Mixed) The design methodology can be generalised to other domains, but the presented system is domain-specific to the Linux Kernel and the out-of-tree patch-set maintenance task. My side-remark below does offer alternatives that are more standard in Software Engineering; however, those too would be specific to the problem domain once designed, requiring adaptation to a new domain.
- W1 The formalism introduced in §4.1 (Δ,Σ,δ,σ,⟨⋅,⋅⟩) does not reduce the cognitive load of trying to understand the Type 1 and Type 2 migrations. The English explanations serve this role better, and the paper could have its clarity improved if the formalism is dropped, as it is not reused in other parts of the paper.
- W2 Perhaps a nit-pick. I personally really like this work, but this seems to be more so an AI4SE paper with a very strong SE flavour. I understand that NeurIPS does allow application papers, so perhaps this paper falls under that rubric.

---

> ### Author Rebuttal · Authors · 2025-07-30
>
> We are so pleased and grateful for your thoughtful feedback. Thank you! We have carefully considered your comments concerning MigGPT's generalization capabilities, symbolic representation, and deprecated system calls, and have provided thorough responses to each of these points. Please refer to the specific responses below and our rebuttal for more information.
>
> ---
>
> **Strength/Weakness Mixed.** Generalization of MigGPT
>
> Thank you for an insightful question about MigGPT's generalizability. We've strengthened our findings with new experiments:
>
> - **MigGPT Excels in Cross-Language Migration:** To directly address concerns about its applicability beyond C-language, we rigorously tested MigGPT on Python code migration. By randomly translating 31 examples from our benchmark and making only minor adjustments for Python's syntax (specifically in tokenization and element extraction), MigGPT achieved a 16.13% average improvement in semantic match performance over traditional methods. This is a powerful testament to its cross-language capabilities.
>
> | LLM | Method | Best Match | Semantic Match |
> | --- | --- | --- | --- |
> | GPT-4-turbo | Vanilla | 61.29% | 70.97% |
> | GPT-4-turbo | MigGPT | 80.65% | 87.10% |
> | DeepSeek-V3 | Vanilla | 64.52 | 74.19% |
> | DeepSeek-V3 | MigGPT | 83.87% | 90.32% |
> - **Proving Versatility Across Migration Challenges:** Our original paper (Table 5) already showcased MigGPT's versatility through its successful application to CVE patch backporting. In this different, yet critical, migration task, MigGPT consistently outperformed existing solutions. In this task, we only need to modify MigGPT's task prompt (to describe the details of the CVE patch backporting task); no modifications are required for the CFP.
>
> ---
>
> **Weakness 1.** Symbol Notation (Response to Weakness 1)
>
> We appreciate the reviewer's insightful feedback. To improve clarity and streamline the presentation, we will simplify ****the expression of this part. Thank you for helping us enhance the readability of our manuscript.
>
> ---
>
> **Weakness 2.** AI4SE (Response to Weakness 2)
>
> Your positive feedback and generous words regarding our work are truly appreciated! We're so pleased it resonated with you and had an impact. We focused on the intersection of AI and software engineering, as we believe this domain offers significant opportunities for practical impact. Our future plans include expanding this work to a broader range of code application areas, and we indeed plan to further delve into the SE aspects of our research and submit to a software engineering conference.
>
> ---
>
> **Question 1.** Deprecated System Calls
>
> Thank you very much for your insightful question. This is indeed a profound point that deserves thorough discussion.
>
> Regarding your question about whether "out-of-tree patch-sets used now deprecated system calls" in our dataset, such instances are rare. This is primarily due to Linux kernel development principles prioritizing backward compatibility. Once a system call is added to the kernel, it's generally supported indefinitely to avoid breaking existing software that relies on it.
>
> However, kernels do evolve, introducing new system calls that offer more robust, secure, or efficient functionalities. These new features can, in some cases, functionally supersede older, more limited mechanisms.
>
> When a patch migration involves new system calls introduced in a newer kernel version, MigGPT is well-equipped to handle such scenarios.  In these cases, the new version target code (s_new) will contain examples of how the new system call is used. MigGPT can then refer to these usage examples in s_new to complete the migration and generate a corresponding patch that aligns with the new API. Certainly, in this scenario, migration types can still be categorized according to the rules in Table 1 of the paper. If the new kernel version and patch modifications don't overlap in location, it's Type 1. Otherwise, it's Type 2.

---

> > ### Comment · Reviewer_Nf59 · 2025-08-01
> >
> > Thank you for the detailed rebuttal.
> >
> > Thank you for validating on Python syntax, and I am happy to see the improvement persists.
> >
> > With regards to the LK, I agree that backwards compatibility is maintained for extreme periods, though eventually there are deprecations (for example, support for, now, exotic hardware). In the context of the paper, I agree that it is reasonable to assume no deprecations, rather co-existence with new syscalls that may subsume or enhance previous syscalls. In this case, I agree; these migrations are either Type 1 or 2. I still think there may be scenarios where features can end up deprecated with no replacements or other fail edge cases, but then it can be argued that the preconditions of the tool do not apply.
> >
> > I maintain my positive rating. I am happy with the work presented in this paper!

---

> > > ### Author Response · Authors · 2025-08-02
> > >
> > > Thank you very much for your positive feedback and for intending to recommend our manuscript for acceptance. We truly appreciate the time and effort you have dedicated to reviewing our work, as well as your valuable insights and constructive comments, which have greatly helped us improve the manuscript. Your support and recognition are sincerely appreciated.

---

### Official Review · Reviewer_igwW · 2025-07-03

**Clarity:** 3
**Significance:** 3
**Originality:** 3
**Rating:** 4
**Confidence:** 4

**Summary:**

This paper introduces MIGGPT, a novel framework designed to automate the migration of out-of-tree Linux kernel patches across different kernel versions, a task that is traditionally manual, time-consuming, and requires significant expertise. The authors identify key challenges LLMs face when applied directly to this task, including misunderstanding incomplete code contexts and inaccurately identifying migration points. To address these issues, MIGGPT reframes the migration process into a two-step task: retrieving target code from the updated kernel and then performing the patch migration.

Authors propose a new data structure called Code Fingerprint (CFP), which is designed to encapsulate the essential structural and critical information of code snippets to provide better context for the LLM. The framework integrates three modules to improve accuracy and efficiency: a Retrieval Augmentation Module, a Retrieval Alignment Module, and a Migration Enhancement Module. The paper also introduces a new benchmark for evaluating out-of-tree kernel patch migration, built from three real-world open-source projects containing 135 migration tasks. The evaluation shows that MIGGPT significantly improves the success rate of patch migration.
ease in computational overhead.

**Questions:**

- Could you provide a more detailed algorithmic description of the Code Fingerprint (CFP) generation process? For example, what specific heuristics or parsing techniques are used to identify the different statement types shown in Figure 4? How does the process handle complex, nested structures or preprocessor macros that are common in kernel code?
- The paper demonstrates impressive results on migrating patches for RT-PREEMPT, Raspberry Pi Linux, and HAOC. How do you see MIGGPT generalizing to other, potentially very different, out-of-tree kernel patches or even to code migration tasks in other large-scale C-based projects? What adaptations to the Code Fingerprint or the prompting strategies might be necessary for different codebases?
- Could you provide a qualitative analysis of cases where MIGGPT fails? Are there common patterns in these failures? For instance, do they relate to particularly complex code refactorings in the kernel, specific types of patch modifications, or limitations in the underlying LLMs that even the CFP cannot overcome? A deeper understanding of the failure modes would be very valuable.

**Ethical Concerns:**

["NO or VERY MINOR ethics concerns only"]

**Final Justification:**

After reading authors responses as well as fellow reviewers feedback, I keep my positive scope of the paper.

**Limitations:**

yes

**Quality:**

2

**Strengths And Weaknesses:**

Strengths
- The paper tackles a highly relevant and challenging real-world problem in software engineering, the maintenance of out-of-tree Linux kernel patches. To best of my knowledge, this is a novel code migration benchmark and is a solid contribution to LLM community.
- The proposed method, MIGGPT, is well-motivated by a clear analysis of the shortcomings of vanilla LLMs for this task. The experimental evaluation is thorough, using a variety of both closed-source and open-source LLMs and demonstrating robust improvements across the board.
- The paper is well-structured and clearly written. The problem is well-defined, and the challenges of using LLMs are articulated with concrete examples referenced in the appendix.

Weaknesses:
- The paper contrasts CFP with Abstract Syntax Trees (ASTs) but a more formal definition and a more detailed, step-by-step example of the CFP generation process would enhance the reader's understanding of this key component.
- Given that LLMs can process sufficiently large context 100k+ tokens, it's not clear what value does CFT abstraction provide against a simple baseline of providing whole file.
- The paper's evaluation of migration success relies on "semantic match" and "human match", with the latter being the more stringent and important metric. While the "human match" results are impressive, the paper could benefit from a more detailed analysis of the cases where MIGGPT fails. Understanding the remaining failure modes would provide valuable insights for future work in this area.
- While the application to out-of-tree kernel patches is novel, the high-level idea of using program analysis techniques to create structured representations of code to guide LLMs is an emerging area of research. The paper could strengthen its contribution by more explicitly situating the Code Fingerprint concept within this broader context of research on combining structured analysis and large language models for code.

-

---

> ### Author Rebuttal · Authors · 2025-07-30
>
> Thank you for your thoughtful feedback. We have carefully considered your comments concerning the generation process of CFP, the generalization of MigGPT, failed cases of MigGPT, and the discussion of CFP. We discussed and supplemented the experiment to address the concern. Please refer to the specific responses below and our rebuttal for more information.
>
> ---
>
> **Weakness 1.** The Generation of CFP
>
> The CFP generation algorithm is detailed in **Algorithm 3 of Appendix D.4**, as mentioned on line 232 of the manuscript. The supplementary materials also include the implementation code for CFP.
>
> In brief, the CFP generation process involves the following steps:
>
> - Tokenization and Scope Identification: The code snippet `s` is first tokenized. We identify nested code scopes (e.g., `{}` and `#ifdef/#endif`) via bracket matching.
> - Code Block Demarcation: Critical symbols (e.g., `{`, `}`, `#ifdef`, `func()`) are used to demarcate different code blocks.
> - Function Call Detection: Function calls are detected through pattern matching on token sequences, for instance, identifying an identifier followed by `()`.
> - Associated Function Identification: We identify associated functions by analyzing call statements within code blocks, avoiding the need to construct complex call graphs.
> - Preprocessor Macros: Preprocessor macros (e.g., `#ifdef/#endif`) are identified via specific keyword matching and treated as distinct code block boundaries.
>
> We cannot provide a step-by-step example figure of the CFP generation process for you due to **the rebuttal policy restrictions ("prohibit using any links in the rebuttal"),** but following your advice, **we will revise our paper to include such a figure**. Thanks for your valuable suggestions.
>
> ---
>
> **Weakness 2.** Baseline
>
> Thank you for your question regarding the context window of LLMs. While LLMs with large contexts can process entire code files, they cannot analyze the structural information needed for code migration, such as target code structure information, migration point information, and patch difference information. As mentioned in Section 5.1 of our paper, there are challenges hindering LLMs’ success in out-of-tree kernel patch migration:
>
> - **Structural Ambiguity**: LLMs struggle to accurately identify function definitions, often getting confused by similar structures.
> - **Boundary Indeterminacy**: Inherent randomness in LLM-generated responses leads to inconsistencies between LLM-identified and human-retrieved target code snippet boundaries.
> - **Missing Associated Fragments**: LLMs often overlook critical code fragments relevant to the migration, resulting in migration failures.
> - **Ambiguous Migration Points**: LLMs cannot precisely identify and locate code migration points.
>
> Our experimental results strongly demonstrate the necessity of CFP.
>
> The baseline vanilla (”two-step strategy”) approach in Tables 2 and 3 is completely equivalent to MigGPT in terms of the context provided to the LLM, as both offer "the whole file" context relevant to migration. The only difference is that MigGPT uses CFP for the extraction and analysis of migration information, while “two-step” does not use CFP. The experimental data clearly shows that MigGPT improved the average migration accuracy by **45.92%** compared to vanilla. This fully demonstrates the indispensable value of CFP abstraction in code migration tasks.
>
> ---
>
> **Weakness 3.** Failure Cases
>
> We agree that understanding these remaining failure modes is crucial for future advancements.
>
> We've conducted an in-depth analysis of the "human match" errors and identified the following primary categories:
>
> - **Incomplete `s_new` Retrieval for Large Codebases (31.03%):** In some instances, when retrieving `s_new` from `new.c`, the target code `s_new` was significantly large (exceeding 150 lines), leading to incomplete retrieval. This could be due to LLMs' tendency to shift attention when dealing with long contexts.
> - **Deviation from the Migration Point in `s'_new` Generation (24.14%):** Even with precise migration point information provided by CFP, the LLM occasionally failed to strictly adhere to these locations during `s'_new` generation. While minor offsets often didn't impact functionality, they sometimes led to functional errors in code with complex control or data flows. This behavior appears to be related to the inherent randomness of LLMs.
> - **Difficulty in Fusing Divergent Changes in Type 2 Migrations (27.59%):** For certain Type 2 migrations, significant differences between `s_new` and `s_old`'s modifications to `s_old` prevented the LLM from correctly integrating these changes, resulting in errors. This limitation points to challenges related to the LLM's code comprehension and manipulation capabilities.
> - **Miscellaneous (17.24%)**: Errors in symbols, statements, etc., appearing in `s'_new`, such as incorrectly writing `verbose(env, off, size, reg's id)` instead of `verbose(env, off, size, reg->id)`. This may be related to LLM hallucinations.
>
> **It's important to note that these identified failure modes primarily arise from the inherent limitations of the LLMs themselves, rather than architectural flaws within the MigGPT framework.** We will incorporate this detailed failure analysis into the final version of our paper. Thank you again for your insightful feedback.
>
> ---
>
> **Weakness 4.** Structured Analysis + LLMs
>
> Thank you for your valuable feedback.  We agree that explicitly situating our Code Fingerprint concept within the broader context of combining structured analysis and LLMs for code will strengthen the paper's contribution.
>
> We have compared MigGPT with existing research in this emerging area.  For instance, works such as [r-1, r-2] utilize heuristic tree structures to assist LLMs in optimizing kernel configurations during operating system deployment.  Similarly, LLM-based fuzzing tools like [r-3, r-4, r-5] employ formal templates to guide LLMs in generating more effective test cases.  These approaches, much like MigGPT, leverage deterministic information such as trees, graphs, and lists to mitigate the inherent uncertainty and randomness of LLMs, thereby improving performance across various application tasks.  However, it's worth noting that the specific deterministic information used and its data structure are often tailored to the particular task at hand.
>
> We will incorporate this discussion into the Related Work section of the final version of our paper.
>
> [r-1] Chen H, Wen Y, Cheng L, et al. AutoOS: Make your os more powerful by exploiting large language models, Forty-first International Conference on Machine Learning. 2024.
>
> [r-2] Lin H, Li Y, Luo H, et al. BYOS: Knowledge-driven Large Language Models Bring Your Own Operating System More Excellent. arXiv preprint arXiv:2503.09663, 2025.
>
> [r-3] Xu H, Ma W, Zhou T, et al. CKGFuzzer: LLM-Based Fuzz Driver Generation Enhanced By Code Knowledge Graph, 2025 IEEE/ACM 47th International Conference on Software Engineering: Companion Proceedings (ICSE-Companion). IEEE, 2025: 243-254.
>
> [r-4] J. Hu, Q. Zhang, and H. Yin, “Augmenting greybox fuzzing with generative AI,” arXiv preprint arXiv:2306.06782, 2023.
>
> [r-5] Z. Liu, C. Chen, J. Wang, M. Chen, B. Wu, X. Che, D. Wang, and Q. Wang, “Testing the limits: Unusual text inputs generation for mobile app crash detection with large language model,” arXiv preprint arXiv:2310.15657, 2023.
>
> ---
>
> **Question 1.** The Generation of CFP
>
> Please refer to our response to Weakness 1
>
> ---
>
> **Question 2.** Generalization of MigGPT
>
> Thank you for your excellent question regarding MigGPT's generalizability across other tasks and domains. We've conducted new experiments to address this:
>
> - **Broad Migration Task Applicability:** Our initial submission (Table 5) already demonstrated MigGPT's versatility by successfully applying it to CVE patch backporting—migrating patches to older code versions. In this scenario, MigGPT significantly outperformed existing methods. In this task, we only need to modify MigGPT's task prompt (to describe the details of the CVE patch backporting task); no modifications are required for the CFP.
> - **Cross-Language Adaptability:** We tested MigGPT's performance with Python code migration. After randomly translating 31 benchmark examples to Python and making minor adjustments to handle Python syntax (specifically in the statement tokenization and element extraction functions), MigGPT showed an average semantic match improvement of 16.13% over a vanilla approach. This confirms MigGPT's effectiveness beyond C-language code.
>
> | LLM | Method | Best Match | Semantic Match |
> | --- | --- | --- | --- |
> | GPT-4-turbo | Vanilla | 61.29% | 70.97% |
> | GPT-4-turbo | MigGPT | 80.65% | 87.10% |
> | DeepSeek-V3 | Vanilla | 64.52 | 74.19% |
> | DeepSeek-V3 | MigGPT | 83.87% | 90.32% |
>
> ---
>
> **Question 3.** Failure Cases
>
> Please refer to our response to Weakness 3

---

> > ### Comment · Reviewer_igwW · 2025-08-03
> >
> > Thank you for responding to all of my questions. In particular, failure mode analysis is insightful. On "Deviation from the Migration Point" issue, can it be addressed by revising the prompt?

---

> > > ### Author Response · Authors · 2025-08-03
> > >
> > > Thank you for the detailed review and valuable suggestions.  We agree that the "Deviation from the Migration Point" issue is a critical challenge.  Our CFP approach addresses this by extracting precise location information and integrating it into a dedicated “Location Prompt”.  Despite this, we observed that LLMs still failed to strictly adhere to these locations in 24.14% of cases.
> > >
> > > We attribute this to a fundamental challenge: the context window limitations of LLMs.  The Linux kernel is a vast and complex codebase, comprising over 40 million lines of code.  No current LLM can fit the entire codebase into its context window.  Consequently, even with precise migration points, the model lacks complete global control and data flow information, which can lead to logical errors and deviations during the migration process.  These failure cases are not just limitations;  they are crucial insights that highlight key research directions for the future of code migration.  To address this, we plan to explore mechanisms for summarizing code control and data flow.  This would allow us to capture critical context without requiring a massive context window, which will be the main focus of our future work.

---

### Official Review · Reviewer_2EZF · 2025-07-04

**Clarity:** 3
**Significance:** 3
**Originality:** 3
**Rating:** 5
**Confidence:** 4

**Summary:**

This paper introduces MIGGPT, a novel framework designed to automate the challenging task of migrating out-of-tree Linux kernel patches across different kernel versions. The authors identify key failures of vanilla LLMs on this task and propose a multi-stage approach that uses a custom "Code Fingerprint" (CFP) representation and three specialized modules to improve code retrieval and patch migration. The framework is evaluated on a new, real-world benchmark, demonstrating significant performance gains over various state-of-the-art LLMs.

**Questions:**

1. Can more migration scenarios be added, such as complex cases where patches involve changes in driver interfaces and kernel subsystem refactoring?​

2. When the target kernel version is quite different from the original version (such as spanning 5 major versions), will the performance of MIGGPT decline significantly? Is there an incremental migration strategy?​

**Ethical Concerns:**

["NO or VERY MINOR ethics concerns only"]

**Final Justification:**

After carefully reviewing the authors’ rebuttal and experiments on cross-language generalization, the fair “one-step + CFP” vs. “two-step + CFP” comparison, migration scenarios, and multi-version migration strategy, I am more convinced by the technical rigor and theoretical completeness of MIGGPT. I therefore raise my recommendation to **5: Accept**.

**Limitations:**

yes

**Quality:**

3

**Strengths And Weaknesses:**

**Strengths**
1. The paper tackles a highly practical and challenging real-world problem in software engineering. The creation of a new, robust benchmark using patches from well-known projects like RT-PREEMPT and Raspberry Pi Linux is a significant contribution that will benefit future research.
2. The proposed Code Fingerprint (CFP) is an elegant and well-motivated solution. It pragmatically avoids the rigidity of traditional Intermediate Representations (like ASTs) by handling non-compilable code and preserving kernel-specific information like comments and inline assembly, which are crucial for this task.
3. The evaluation is comprehensive and convincing. The authors test their framework against a wide array of modern LLMs, employ multiple evaluation metrics including human judgment, and provide a thorough ablation study that validates the contribution of each component of their system.

**Weakness**
1. The proposed MIGGPT framework is quite complex and heavily engineered for the specific task of Linux kernel patch migration. While effective, its generalizability to other code migration tasks (e.g., in different programming languages) is unclear.

2. The comparison between the proposed "two-step strategy" and the "one-step strategy" may not be entirely fair. The one-step baseline appears to be a naive, direct query to the LLM. A stronger baseline might involve a more sophisticated one-step prompt that incorporates some of the contextual cues used by MIGGPT, which could provide a more nuanced measure of the framework's benefit.

---

> ### Author Rebuttal · Authors · 2025-07-30
>
> Thanks for your valuable feedback. We've addressed your comments on MigGPT's generalization, the fairness of the "one-step strategy" comparison, additional migration scenarios, and cross-major-version migration. We discussed and supplemented the experiment to address the concern. Please refer to the specific responses below and our rebuttal for more information.
>
> ---
>
> **Weakness 1**. Generalization of the Framework
>
> We thank you for raising an important question about the generalizability of MigGPT to other programming languages migration scenarios or other domains. Below, we have added a new experiment:
>
> - Cross-Language Generalizability: To address the concern about **cross-language generalizability**, we conducted an additional experiment focusing on Python code migration. We randomly selected 31 migration examples from our benchmark and translated them to Python. With only minor adjustments (modify the implementation of the statement tokenization and statement element extraction functions) to adapt the CFP for Python syntax, MigGPT was able to process these examples. The results demonstrate that the average semantic match performance improved by 16.13%. compared to a vanilla approach, indicating MigGPT's effectiveness beyond C-language code.
>
> | LLM | Method | Best Match | Semantic Match |
> | --- | --- | --- | --- |
> | GPT-4-turbo | Vanilla | 61.29% | 70.97% |
> | GPT-4-turbo | MigGPT | 80.65% | 87.10% |
> | DeepSeek-V3 | Vanilla | 64.52 | 74.19% |
> | DeepSeek-V3 | MigGPT | 83.87% | 90.32% |
> - Generalizability to Other Migration Tasks: Our original submission already showcased MigGPT's applicability to **other code migration tasks**. As presented in **Table 5** of our paper, MigGPT was successfully applied to the **CVE patch backporting** task (migrating a CVE patch from the version where the CVE was discovered to an older version). In this task, we only need to modify MigGPT's task prompt (to describe the details of the CVE patch backporting task); no modifications are required for the CFP. The results demonstrate MigGPT's superior performance compared to existing methods in this different migration scenario.
>
> ---
>
> **Weakness 2.** Misunderstanding of the "One-step Strategy”
>
> Apologize for the confusion, but we want to clarify that the contextual information given to the LLM is **identical for both the "two-step strategy"** (baseline vanilla approach in Tables 2 and 3)  **and the "one-step strategy", so the comparison is fair** (details are shown in the table below). The contextual information includes the **old version code** `s_old` , `s'_old` and the **new version file** `new.c`. As shown in Figure 5 of the paper, the "two-step strategy" is clearly superior to the "one-step strategy".
>
> | Context | One-step strategy | Two-step strategy (Vanilla) | One-step strategy + CFP | MigGPT (Two-step strategy + CFP) |
> | --- | --- | --- | --- | --- |
> | Code Information | ✔ | ✔ | ✔ | ✔ |
> | CFP Information | ✘ | ✘ | ✔ | ✔ |
>
> To further address your concerns, we provided the key migration information extracted by CFP and the code information (`s_old` , `s'_old`, `new.c`) as context to the "one-step strategy" (One-step + CFP) and compared it with MigGPT (Two-step + CFP). As shown in the table below, formulating the migration task as a two-step process demonstrates performance advantages.
>
> |  | Best Match | Semantic Match |
> | --- | --- | --- |
> | One-step + CFP | 38.52% | 42.96% |
> | MigGPT | 62.96% | 80.00% |
>
> We recognize we didn't make this clear enough in the current paper, which likely led to this misunderstanding. We'll **revise the manuscript to explicitly emphasize this alignment of contextual information** for both strategies, ensuring readers fully understand the fairness of our comparison and our framework's true impact. Thanks for your valuable comment.
>
> ---
>
> **Question 1.** More Migration Scenarios
>
> We appreciate the reviewer's insightful question regarding more complex migration scenarios. Kernel subsystem refactoring: Indeed, our experiments already encompass kernel subsystem refactoring. For instance, our benchmarks include significant memory management subsystem changes, such as the introduction of folio and enhancements to io_uring. We also cover the process management subsystem's shift from CFS to EEVDF. These cases demonstrate our approach's effectiveness in handling substantial internal kernel restructuring.
>
> Driver interfaces: While our paper primarily focuses on out-of-tree patch migration, we've conducted supplementary experiments to assess MigGPT's performance on these cases and address your concerns. We randomly collected 24 driver migration samples. The results, presented in the table below, indicate that MigGPT performs well in migrating patches involving driver interface modifications.
>
> | LLM | Method | Best Match | Semantic Match |
> | --- | --- | --- | --- |
> | GPT-4-turbo | Vanilla | 54.16% | 62.5% |
> | GPT-4-turbo | MigGPT | 83.33% | 83.33% |
> | DeepSeek-V3 | Vanilla | 58.33% | 62.5% |
> | DeepSeek-V3 | MigGPT | 79.16% | 87.5% |
>
> ---
>
> **Question 2.** Migrating Across Multiple Versions
>
> Our benchmarks have already included the scenario where the target kernel version is quite different from the original. For instance, we demonstrate MigGPT's superior performance in migrating code from Kernel 5.10 to Kernel 6.6. This migration spans 15 versions (5.10, 5.11, …, 5.19, 6.0, 6.1, …, 6.6), clearly showing our tool's effectiveness across substantial version gaps.
>
> Furthermore, Linux kernel development, and especially the maintenance of out-of-tree patches by vendors, typically focuses on Long-Term Support (LTS) versions (e.g., 4.19, 5.4, 5.10, 6.6).   It's uncommon for vendors to directly develop and maintain patches across multiple, non-consecutive LTS versions.  Our benchmarks specifically target migrations between these LTS versions, making them representative of real-world out-of-tree patch migration scenarios.
>
> | Version | 4.19 | 5.4 | 5.10 | 6.6 |
> | --- | --- | --- | --- | --- |
> | Released Time | 2018-11 | 2019-11 | 2020-12 | 2023-10 |
>
> However, if an extreme case necessitates migrating across multiple LTS versions (e.g., from 4.19 to 6.6), an incremental migration strategy can be employed. A vendor could first use MigGPT to migrate from Kernel 5.4 to 5.10, rigorously test the migrated code, and resolve any bugs to ensure functionality.  Subsequently, MigGPT could then be used for the migration from Kernel 5.10 to 6.6.

---

> > ### Comment · Reviewer_2EZF · 2025-08-05
> > **Thank you for your detailed responses!**
> >
> > Thank you for your detailed responses to the concerns I raised. I hope you will incorporate these improvements into the paper, as they will greatly enhance its validity. Consequently, I have increased my scores.

---

> > > ### Author Response · Authors · 2025-08-06
> > >
> > > Thank you very much for your positive feedback and encouragement. We are very grateful to hear that you are satisfied with our responses and have increased your scores. We will ensure that your valuable suggestions are fully incorporated into the paper to improve its quality.

---

### Note · Authors · 2025-08-12

Dear ACs and reviewers:

We thank the reviewers for their valuable comments and suggestions on our paper. We believe that we have adequately addressed most of the major concerns during the rebuttal phase. We are particularly pleased to note that one reviewer has indicated a willingness to raise their score.

The reviewers agree that,

(1) We have addressed their concerns on the generalization of the framework (2EZF, igwW, Nf59, NY3E), method, and experimental details (2EZF, igwW, Nf59, NY3E) and future directions for the framework (igwW, Nf59).

(2) Our work is novel and practical, as it tackles a highly challenging real-world problem in Linux kernel maintenance (2EZF, igwW). Our creation of a new code migration benchmark is a major contribution (2EZF, igwW), and our proposed methods are well-motivated solutions (2EZF, igwW).

(3) Our experiments are a comprehensive design (2EZF, igwW). They cover a wide range of LLMs (2EZF, igwW) and utilize diverse evaluation metrics (2EZF). Our results are impressive and strong (NY3E), which demonstrates the effectiveness of our framework.

We believe that the quality of our paper has been significantly improved through this revision and discussion. We look forward to the final decision.

Sincerely,

Authors

---

### Decision · Program_Chairs · 2025-09-17

**Decision:**

Accept (spotlight)

**Comment:**

MigGPT presents a technically solid and impactful contribution to the intersection of LLMs and software maintenance, offering SOTA results on a difficult and valuable task. The "Code Fingerprint" representation, is an elegant and well-motivated solution.

Despite concerns about generalizability and baseline comparisons, the authors have adequately addressed these issues in the rebuttal, and none of the identified weaknesses undermine the central contributions or experimental findings. Given the strong real-world relevance, novel methodology, and robust results, I recommend acceptance.